# Maternal histone methyltransferases antagonistically regulate autosomal random monoallelic expression (aRMAE) in C. elegans

Bryan Sands, Soo R. Yun, Junko Oshima & Alexander R. Mendenhall ⬛ ✉

Undefined epigenetic programs act to probabilistically silence individual autosomal alleles, generating unique individuals, even from genetic clones. This random monoallelic expression can explain variation in traits and diseases that differences in genes and environments cannot. Here, we developed the nematode *Caenorhabditis elegans* to study monoallelic expression in whole tissues, and defined a developmental genetic regulation pathway. We found maternal H3K9 histone methyltransferase (HMT) SET-25/SUV39/G9a works with HPL-2/HP1 and LIN-61/L3MBTL2 to randomly silence alleles in the intestinal progenitor E-cell of 8-cell embryos to cause monoallelic expression. SET-25 was antagonized by another maternal H3K9 HMT, MET-2/SETDB1, which works with LIN-65/ATF7IP and ARLE-14/ARL14EP to prevent monoallelic expression. The HMT catalytic SET domains of both MET-2 and SET-25 were required for regulating monoallelic expression. Our data support a model wherein SET-25 and MET-2 regulate histones during development to generate patterns of somatic monoallelic expression that are persistent but not heritable.

When accounting for variation in traits, scientists bin variation into genes, environment, and a third component that accounts for biological variation that is considered nongenetic and nonenvironmental[1]. In model animal research, it is possible to hold genes and environments as constants, so the remaining variation in traits accounts for this nongenetic, nonenvironmental variation, which can be substantial[2]. Terms like incomplete penetrance and variable expressivity were coined to explain the observed variation in mutant phenotypes in isogenic populations of flies in homogeneous environments[3–5]. This intrinsic variation in discrete or complex traits can manifest through intrinsic variation in gene expression[6,7], including through the endogenous silencing of genes, or even through the silencing of individual alleles[8–10]. This can account for incomplete penetrance, missing heritability, and nongenetic, nonenvironmental variation in complex traits, including the manifestation of chronic diseases. We refer to this probabilistic epigenetic silencing of alleles as autosomal random monoallelic expression (aRMAE), reviewed in refs. 8–10.

## Monoallelic expression in humans

aRMAE is widespread[11] and consequential, but not entirely understood[8–10]. Since it is propagated through tissues, it has been detectable with both bulk RNA-seq[12,13] and ChIP-seq[14,15]. Recent analysis of GTEX RNA-seq data has shown that genes with aRMAE potential are overwhelmingly enriched for effects on aging and chronic disease like cancer[13]. Moreover, aRMAE operates on a continuum, from fully monoallelic to biallelic, and the silencing of alleles is persistent, but not heritable between generations[9,16].

Monoallelic expression affects human genetic disease and cancer. There are several cases of monoallelic expression affecting the penetrance of genetic disease, cited in these reviews[8–10,17]. One example is a case study of a family that harbored a dominant-negative mutation in *PIT1*[18]. In the three heterozygous individuals that harbored the dominant-negative disease allele (grandmother, father, and daughter), only the daughter was affected by disease. Analysis of RNA showed that the pathological disease allele was silenced in the disease-free family members (grandmother and

Department of Laboratory Medicine and Pathology, School of Medicine, University of Washington, Seattle, WA, USA. ✉e-mail: alexworm@uw.edu

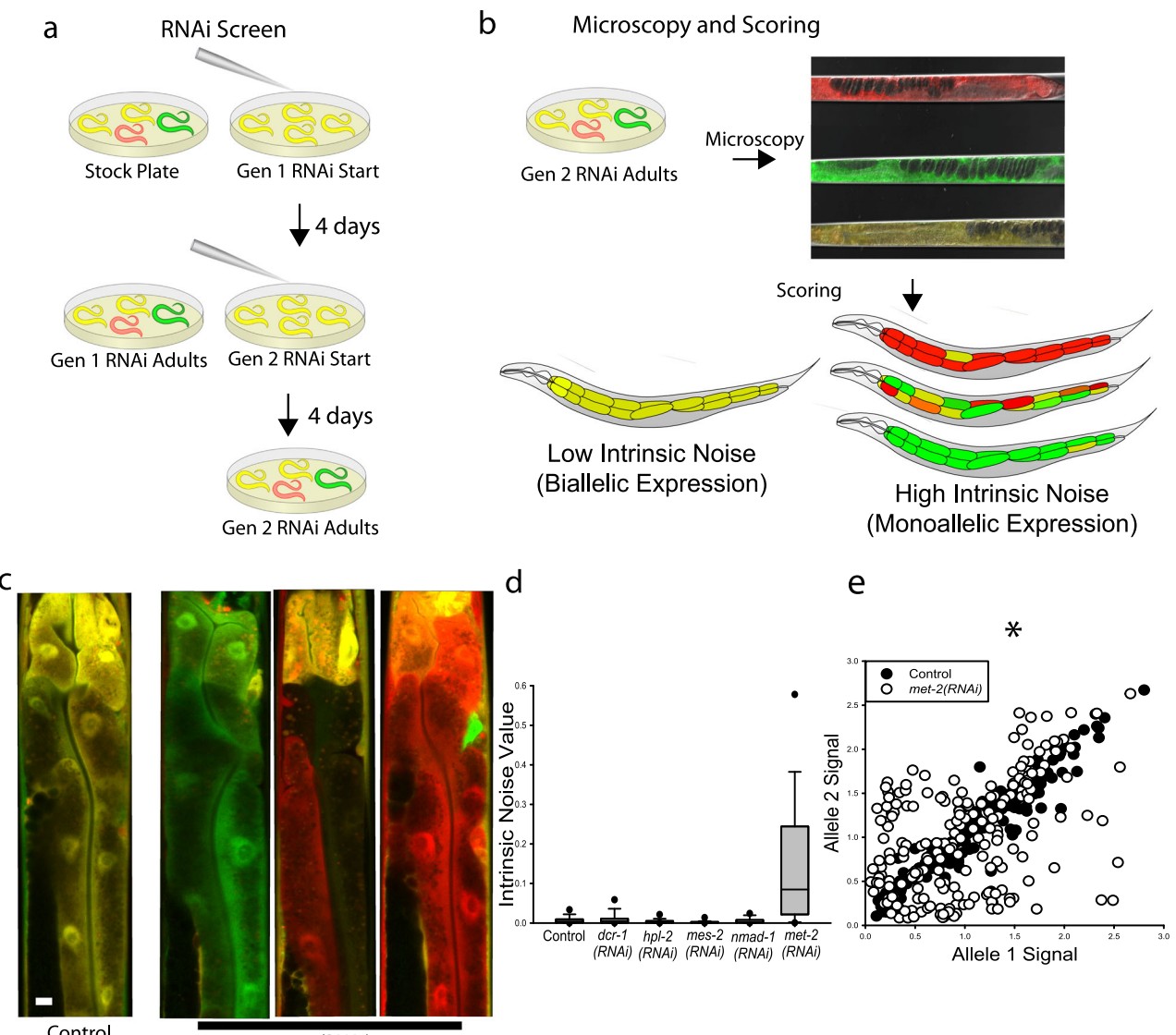

**Fig. 1 | MET-2 negatively regulates monoallelic expression. a** schematic overview of how we implemented 2 generations of RNAi in animals that were heterozygous for reporter alleles. **b** animals were mounted in microfluidic devices for live imaging by confocal microscopy. Cartoons of the spectrum of phenotypes we would observe. **c** merged confocal images of the "torso" of control animals and *met-2(RNAi)* animals. The torso image is the section of the animal that fits into the field of view of our 40× objective and contains cells in the first three or four rings of the intestine. White scale bar on bottom left of control animal is 10 micrometers. **d** data from a small subset of genes in our genetic screen for negative regulators of aRMAE. Boxplots of intrinsic noise, a quantification of allele bias/aRMAE, calculated from the intestine cell allele expression values. Top of boxplot is 75th percentile, bottom of box is 25th percentile, line is median, top and bottom error bars are 90th and 10th percentile, respectively, and dots are 95th and 5th percentile. **e** intestine cells plotted by allele expression level from control and *met-2(RNAi)* animals. There was a significant increase in intrinsic noise for *met-2* animals compared to control animals (0.00371); *P* < 0.05; Kruskal–Wallis One Way Analysis of Variance on Ranks followed by Dunn's Procedure, *N* = 207 cells for control, *N* = 208 cells for *met-2(RNAi)*, three independent experiments. See also Supplementary Information for additional statistical details and Supplemental Fig. 1, showing developmental events indicated by monoallelic expression patterns and detailing additional RNAi test results.

father), and expressed in the affected daughter. In a recent study of inborn errors of immunity, Stewart et al. showed that genes responsible for the penetrance of inborn errors of immunity, such as *PLCG2* and *JAK1* were expressed in a monoallelic fashion in T-cells when inborn errors of immunity were escaped, or in a biallelic fashion when inborn errors of immunity manifested[19].

The variable onset of genetic diseases can also be explained by aRMAE. Genetic diseases like *ACTG2*-related visceral myopathy have variable onset, with some people remaining symptom-free until later in life[20]. This late onset is consistent with the idea that desilencing of pathological alleles could cause the later onset of disease. Moreover, the silencing of non-pathological alleles in people that are heterozygous for recessive pathological alleles can lead to

manifestation of recessive disease that Mendelian genetics would not predict. Finally, in addition to changes in biochemical activity conferred by alleles with different sequence, silencing of alleles will generally lower protein dosage/activity (some genes have compensation mechanisms[21]).

Monoallelic expression of several genes in tumors can act as prognostics in cancer medicine[22–29]. Monoallelic expression can mean increased life expectancy or decreased life expectancy, depending on the particular scenario. In one report, patients with tumors that were heterozygous for *IDH1* gene mutations survived longer than patients with monoallelic expression of the normal *IDH1* allele in tumors[22]. Homologs for genes we identified as regulating aRMAE in this report affect the immunogenicity of tumors[30] and

**Table 1 | Targeted genetic screen for negative regulators of monoallelic expression**

| Gene | Predicted or confirmed function | Coefficient of determination (R²) for alleles |
|---|---|---|
| Empty vector | Normal wild-type functions under ad libitum laboratory conditions | 0.928 (SD = 0.04) |
| set-1 | Predicted HMT | RNAi lethal |
| set-2 | H3K4 HMT, SETD1a | 0.9547 |
| set-3 | Predicted HMT, SMYD4 | 0.9355 |
| set-4 | Predicted HMT, H4K20 specific, KMT5C | 0.8564 |
| set-5 | Predicted HMT, KMT2E/SETD5 | 0.9167 |
| set-6 | Predicted HMT, H3K9? | 0.845 |
| set-7 | ATP binding, TTLL12 | 0.9333 |
| set-8 | SET domain | 0.8237 |
| set-9 | SET domain, set-26 paralog, SETD5, H3K4me3 reader | 0.5827 |
| set-10 | Predicted HMT, distant SYMD1/2 | 0.7147 |
| set-11 | Predicted HMT, p53 binding, EHMT1/2/SUV39H1/H2 | 0.8747 |
| set-12 | Predicted HMT, H3K36 specific, NSD1/2 | 0.9089 |
| set-14 | Predicted HMT, SMYD2 | 0.7095 |
| set-15 | SET domain | 0.8311 |
| set-16 | H3K4 specific HMT, KMT2D | 0.9028 |
| set-18 | H3K36 specific HMT, SMYD3 | 0.8217 |
| set-19 | SET domain | 0.8035 |
| set-20 | SET domain | 0.9422 |
| set-21 | Predicted H3K9 HMT? | 0.9691 |
| set-22 | SET domain | 0.9656 |
| set-24 | SET domain | 0.9451 |
| set-25 | H3K9-specific HMT | 0.963 |
| set-27 | Predicted H4K4 and H3K36 HMT, SETD3 | 0.9312 |
| set-28 | SET domain | 0.9122 |
| set-30 | Predicted H3K4 specific HMT, SMYD1/2/3 | 0.9683 |
| set-31 | SET domain | 0.9083 |
| set-32 | H3K23me3 HMT | 0.9444 |
| hpl-2 | Human HP1, condensed chromatin | 0.9804 |
| met-2 | H3K9me1/2 specific HMT, SETDB1 | 0.2753 |
| mes-2 | H3K27 specific HMT, EZH1/2, polycomb/PRC2, X-inact | 0.98 |
| mes-4 | H3K36 specific HMT, silences repetitive arrays | 0.9364 |
| utx-1 | H3K27 specific demethylase, KDM6A/B UTX/UTY | 0.9206 |
| jmjd1.2 | H3K9me2 demethylase | 0.7343 |
| jmjd-2 | H3K36me3 demethylase | 0.9722 |
| jhdm-1 | H3K36me3 demethylase | 0.9685 |
| spr-5 | H3K4 demethylase, KDM1A | 0.887 |
| wdr-5 | H3K4 HMT, MLL/ set/COMPASS, WDR5 | 0.9233 |
| lsd-1 | H3K4 demethylase, KDM1A | 0.9606 |
| mrg-1 | H3K4me/H4K16ac transcriptional silencing | 0.9453 |
| prg-1 | Piwi RNA-mediated gene silencing, PIWIL1 | 0.8671 |

**Table 1 (continued) | Targeted genetic screen for negative regulators of monoallelic expression**

| Gene | Predicted or confirmed function | Coefficient of determination (R²) for alleles |
|---|---|---|
| hrde-1 | RNAi effector, exo-RNAi targets, RISC complex | 0.9012 |
| ergo-1 | 26 G RNAi pathway, repression of repetitive sequences | 0.9539 |
| nrde-2 | Recruited to nrde-3 complexes, exo-RNAi RISC complex | 0.9788 |
| nrde-3 | RNAi effector, exo-RNAi targets in soma | 0.9452 |
| nrde-4 | Recruited to nrde-3 complexes, exo-RNAi RISC complex | 0.7605 |
| sago-1 | AGO, siRNA amplification exo RNAi | 0.9195 |
| sago-2 | AGO, siRNA amplification exo RNAi | 0.8247 |
| dcr-1 | Required for RNAi, small RNA processing | 0.9196 |
| morc-1 | Endo-siRNA, HUSH complex | 0.8926 |
| nmad-1 | DNA methylase, ALKBH4 | 0.9514 |
| damt-1 | DNA methylase (adenine), METTL4 | 0.8961 |

RNAi Screen for regulators of allele bias. Genes screened, their function, and allele bias score (R². coefficient of determination for all cells in all worms, *n* = 70 cells from 10 worms). Control empty vector RNAi was conducted in triplicate, with mean and +/− SDEV shown.

cancer cell lines[31], possibly through effects on monoallelic expression of immunogenic proteins. Finally, assessment of silencing states opens up treatment options for some patients. Ovarian cancer patients that are found to have wild-type but silenced *BRCA1/2* genes are responsive to PARPi treatments, whereas patients with normally expressed BRCA1/2 do not respond[32].

We have developed *C. elegans* as a model to study aRMAE in vivo. This model utilizes two distinctly colored fluorescent alleles integrated at the same locus to visualize and quantify differences in allele expression at the protein level in individual somatic cells. The system addresses a gap between sequencing based approaches to study monoallelic expression and medical cases of monoallelic expression[33]. Moreover, there are technical advantages to using a nematode model to study aRMAE. These include a transparent soma, the ease of conducting genetic screens, a short life cycle, and a fixed mitotic lineage. Together, these properties allow us to directly visualization the expression of individual alleles in multiple tissues, assess heritability, distinguish between steady-state expression and transcriptional bursting, and to infer developmental events from somatic expression patterns[34–41]. We note that, unlike probe or sequencing-based approaches, our approach with fluorescent reporter alleles does not quantify the transcripts of each allele of a gene, but instead quantifies the resulting protein products of each allele.

In this work, we perform a targeted screen for negative regulators of aRMAE and identify the H3K9-specific histone methyltransferase (HMT) *met-2/SETDB1*[42]. A suppressor screen reveals that another H3K9-specific HMT, SET-25[42–44], antagonistically regulates aRMAE with MET-2, similar to the effects that these genes have on germline immortality[45]. Loss of MET-2, or just its catalytic HMT SET domain, promotes aRMAE, while loss of SET-25 activity, or just its SET domain, prevents aRMAE, resulting in biallelic expression (BAE). *met-2* mutant animals show a pattern of monoallelic expression that is persistent throughout adulthood, but not heritable. Additionally, animals lacking functional catalytic domains of MET-2, SET-25, or both, are able to use RNA interference to silence somatic genes,

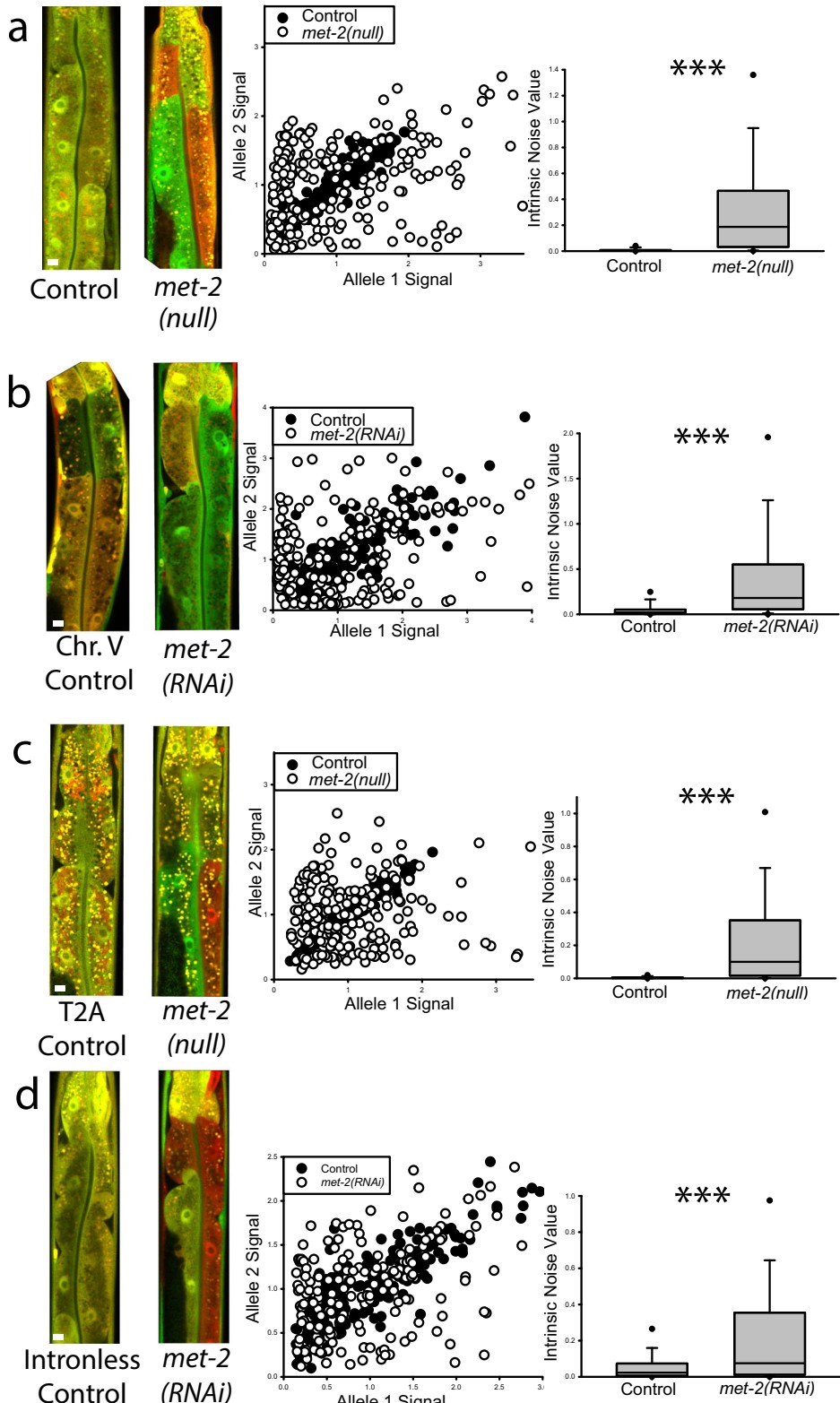

demonstrating that the genes regulating aRMAE are not required for regulating RNA interference. Reciprocal cross experiments show that maternal MET-2 and SET-25 act in the E-cell of the 8-cell embryo to regulate aRMAE. Our results support a model wherein MET-2 and SET-25 competitively regulate histone marks to control nucleosome occupancy and silencing of alleles of genes that are subject to aRMAE.

## Results

### Targeted screening identifies MET-2/SETDB1 as a negative regulator of monoallelic expression

We have previously developed a reporter allele system to quantify expression of alleles at the protein level in single cells of whole tissues in *C. elegans*[34]. Our system utilizes two differently colored fluorescent alleles, such as *hsp-90* reporter alleles, that are expressed

**Fig. 2 | MET-2 regulates monoallelic expression when coding sequence and locus are changed.** For each row, **a**–**d**, left panel "torso" images of control and experimental worms with a 10 micrometer white scale bar inset in the bottom left of each control animal; middle panel scatter plots of control and experimental group intestine cells plotted by allele expression values from all three independent experiments; right panel boxplots of intrinsic noise values from cells from control and experimental worms. Top of boxplot is 75th percentile, bottom of box is 25th percentile, line is median, top and bottom error bars are 90th and 10th percentile, respectively, and dots are 95th and 5th percentile. **a** Intestine cells from *met-2(null)* animals had a significantly higher median intrinsic noise (0.187 compared to 0.00232 for control animals); *P* < 0.001, Mann–Whitney Rank Sum Test, two-sided, *N* = 209 cells for control, *N* = 208 cells for *met-2(null)*, three independent experiments. **b** Worms have reporter alleles inserted at a Chr. V locus. Intestine cells from *met-2(RNAi)* animals had a significantly higher median intrinsic noise (0.180 vs control 0.0196), *P* < 0.001, Mann–Whitney Rank Sum Test, two-sided, *N* = 207 cells for control, *N* = 206 cells for *met-2(RNAi)*, three independent experiments. **c** Worms have full length *hsp-90* reporter alleles inserted at the same Chr. II locus as in (**a**). Intestine cells from *met-2(RNAi)* animals had a significantly higher median intrinsic noise (0.100 vs 0.00163 for control animals), *P* < 0.001, Mann–Whitney Rank Sum Test, two-sided, *N* = 196 cells for control, *N* = 210 cells for *met-2(null)*, three independent experiments.

from the same locus in somatic cells[34]. The animals are propagated by picking self-fertile hermaphrodites that are heterozygous for the reporter alleles at each generation. Under standard conditions (OP50-seeded NGM at 20°) intestine cells express *hsp-90* reporter alleles in a biallelic fashion, with occasional instances of allele bias or aRMAE detected[34].

Here, we conducted a targeted reverse genetic screen to identify negative regulators of aRMAE. To do this, we subjected worms that are heterozygous for *hsp-90* reporter alleles to two generations of RNAi (Fig. 1a), loaded them into microfluidics devices, and then imaged the animals by confocal microscopy (Fig. 1b). Worms expressing *hsp-90* reporter alleles on empty vector (EV) were used as a control. We screened chromatin-modifying genes[46], including SET-domain proteins, DNA modifiers, known or predicted histone modifiers, and proteins associated with the endogenous silencing machinery, for increases in aRMAE (Table 1). In our screen, only *met-2* showed extreme aRMAE, with *met-2(RNAi)* animals showing a pattern of monoallelic expression (strong green or strong red fluorescence only) across most somatic cells, including those that we quantified in the intestine (Fig. 1c–e). *met-2* encodes a conserved H3K9 HMT that is homologous to human *SETDB1*[42–44] and is generally associated with silencing, not preventing it. Figure 1d shows the intrinsic noise values for control and *met-2(RNAi)* compared to a subset of other genes from the screen (full list of genes in Table 1).

To validate the initial *met-2(RNAi)* screen result, we conducted two additional biological replicates of *met-2* RNAi and found that knockdown of *met-2* resulted in a large, significant increase in monoallelic expression, quantified as intrinsic noise (Fig. 1e and Supplemental Fig. 1). Scatter plots of individual intestine cells plotted by allele expression values from control and *met-2(RNAi)* animals are shown in Fig. 1e.

To determine if a null mutant would recapitulate the RNAi phenotype, we created a new *met-2* null allele using CRISPR/Cas9 to insert multiple stop codons into the first exon of the *met-2* reading frame (see "Methods" and Supplemental Tables 1 and 2). The *met-2(null)* animals phenocopied the *met-2(RNAi)*, resulting in significantly higher median intrinsic noise, or significantly increased aRMAE, compared to the control animals (Fig. 2a). To further validate this finding, we conducted *met-2* RNAi on animals expressing the same reporter alleles inserted at a different, noisier locus on Chr. V[34]. Similar to what we found with reporter alleles on Chr. II, *met-2(RNAi)* animals expressing the reporter alleles from Chr. V had a significant increase in aRMAE compared to EV controls (Fig. 2b). These results show that *met-2* negatively regulates aRMAE from two different chromosomal loci that are known to have different baseline aRMAE potential[34].

In prior work, we found that changes to coding sequences did not affect aRMAE, but intron status of alleles did[34]. For example, full length *hsp-90T2A* reporter alleles that contained the full coding sequence and introns showed more BAE compared to intronless versions of the otherwise same alleles[34]. To determine if either coding sequence or intron status was important for *met-2* to function as a negative regulator of aRMAE, we measured the full length *hsp-90T2A* alleles (different coding sequence than the promoter fusions) and *hsp-90* reporter alleles without introns in wild-type and *met-2(null)* genetic

backgrounds. In *met-2(null)* worms expressing the additional full length *hsp-90* coding sequence, intrinsic noise was significantly higher than in controls, shown in Fig. 2c. Similarly, intronless *hsp-90* reporter alleles expressed in *met-2(null)* animals showed significantly higher intrinsic noise than controls, shown in Fig. 2d. Taken together, we validated our initial screen result by determining that *met-2* negatively regulates aRMAE whether the reporters were inserted at different loci, had different coding sequences, or if the coding sequence contained introns or not.

## MET-2 regulates MAE in a gene-specific fashion

We previously found that aRMAE was regulated by promoter, and not influenced by coding sequence[34], consistent with bioinformatic findings[47], and indicative of gene-specific regulation of MAE. In humans, there are at least two distinct pathways controlling MAE, with one utilizing 5mC at promoter regions[12] and the other using H3K9me3 marks at promoter regions[21]. In fact, many studies have found that different monoallelically expressed genes have different epigenetic marks associated with their regulation[12,14,15,21,48]. Thus, there are gene-specific regulatory mechanisms controlling aRMAE. To test the hypothesis that MET-2 regulates aRMAE in a gene or promoter-specific fashion, we built additional reporter alleles and tested them for response to *met-2* RNAi or *met-2* knockout. We examined four distinct reporter alleles. The *vit-2* promoter drives extremely high expression of yolk protein during egg production[49,50]. The *hsp-16.2* promoter controls expression of a small heat shock protein[51,52]. The *idh-1* gene is associated with differences in survival time for patients with brain tumors[22]. Finally, the *eef-1A.1* promoter controls expression of a highly conserved translation elongation factor, also involved in the heat shock response[53].

We found no significant difference in median intrinsic noise between *vit-2* reporter alleles expressed in animals with and without the *met-2(null)* mutation, shown in Fig. 3a. While *hsp-16.2* and *idh-1* have been shown to be expressed in a monoallelic fashion in worms and brain tumors[22,34,37,38], respectively, we found no significant difference in median intrinsic noise for *hsp-16.2* or *idh-1* reporter alleles between control animals and animals on *met-2(RNAi)*, shown in Fig. 3b, c. In contrast, we did see a significant difference in median intrinsic noise from strains harboring *eef-1A.1* reporter alleles on *met-(RNAi)*, shown in Fig. 3d. These results show that *met-2* regulates MAE in a gene-specific manner.

## MET-2 associated genes regulate MAE in unexpected ways

MET-2 is known to work with several other genes to regulate heterochromatin localization and silencing of genes, repetitive elements, and transposons[42,43,54,55]. In worms, MET-2 and SET-25 are responsible for H3K9 methylation through two distinct regulatory pathways[42,43]. In a heterochromatin regulation pathway[54,55], MET-2 adds H3K9m1/2[34], and SET-25 adds H3K9me3[42], with the help of LIN-61[43]. This pathway also requires LIN-65 to bind MET-2 in the cytoplasm and translocate it to the nucleus, and an additional cofactor, ARLE-14, a conserved GTPase effector, whose role is not completely understood[55,56]. Constitutive heterochromatin is tethered to the nuclear periphery

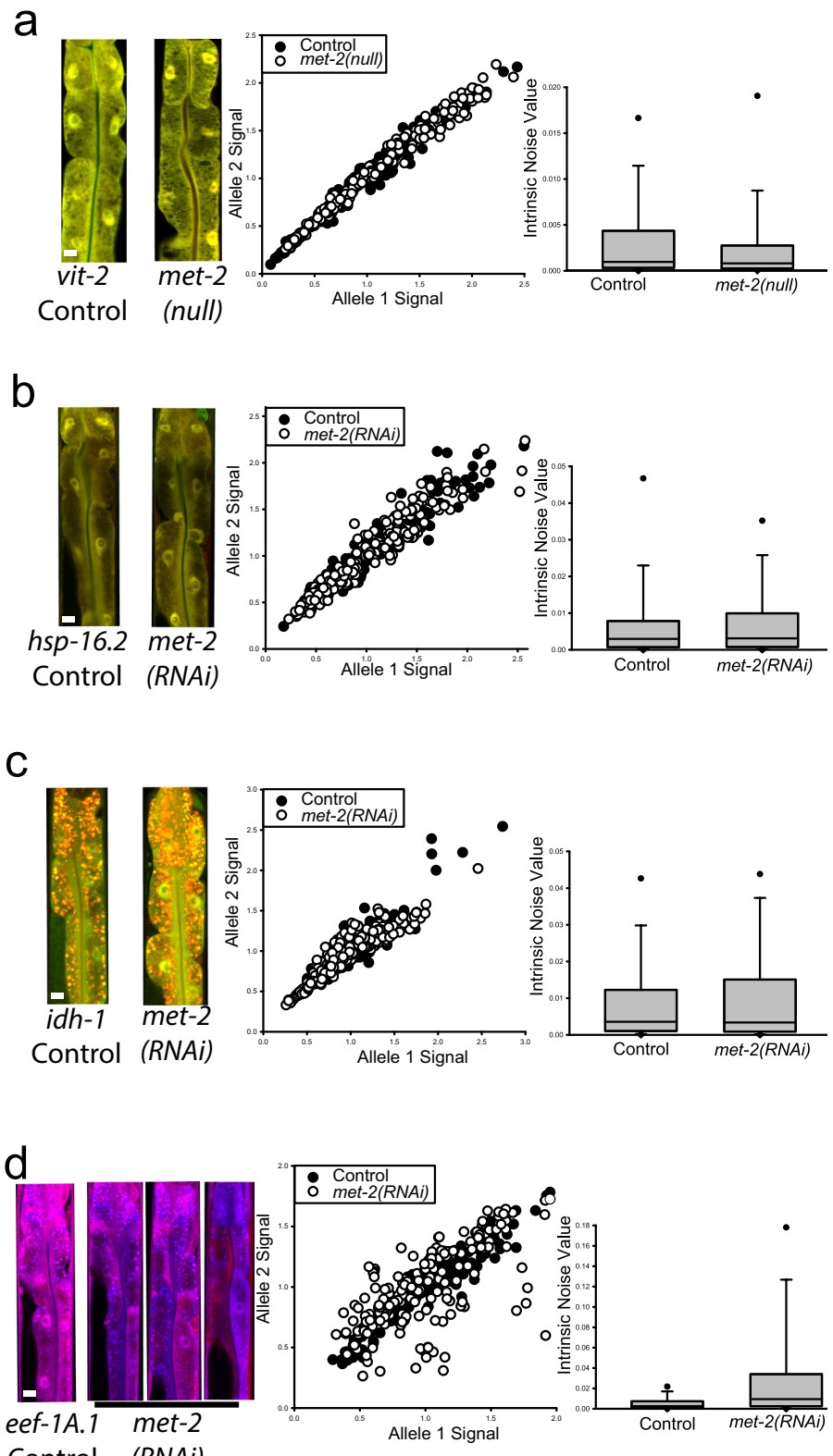

by CEC-4 or LEM-2[57]. In the distinct gene and transposon silencing pathway, SET-25 directly adds H3K9me1/2/3 with the help of *nrde-3*[43].

To determine the role of H3K9me HMTs and their associated cofactors in regulating aRMAE, we conducted RNAi knockdown in the *hsp-90* reporter allele strain and quantified intrinsic noise. Knockdown of *cec-4* or *lem-2* had no effect on aRMAE (Fig. 4a, b). Knockdown of

either *arle-14* or *lin-65* caused a significant increase in aRMAE compared to controls, though the *arle-14(RNAi)* animals showed a subtler phenotype, consistent with prior findings[55,56] (Fig. 4c). The phenotype of *lin-65(RNAi)* animals (Fig. 4d), was visually indistinguishable from *met-2(null)* and *met-2(RNAi)* animals (Figs. 1 and 2). For *lin-61* and *set-25*, the results were the exact opposite of *met-2*, *lin-65*, and *arle-14*. Both

**Fig. 3 | MET-2 negatively regulates monoallelic expression in a promoter/gene-specific fashion.** For each row, **a–d**, left panel "torso" images of control and experimental worms with a 10 micrometer white scale bar inset in the bottom left of each control animal; middle panel scatter plots of control and experimental group intestine cells plotted by allele expression values from all three independent experiments; right panel boxplots of intrinsic noise values from cells from control and experimental worms. Top of boxplot is 75th percentile, bottom of box is 25th percentile, line is median, top and bottom error bars are 90th and 10th percentile, respectively, and dots are 95th and 5th percentile. **a** no significant difference in allele bias for *vit-2* reporter allele expression between *met-2(null)* animals and controls (0.000809 vs 0.000968 control), $P = 0.266$, Mann–Whitney Rank Sum Test, two-sided, $N = 210$ cells for control, $N = 209$ cells for *met-2(null)*, three independent experiments. **b** no significant difference in allele bias for *hsp-16.2* reporter allele expression between *met-2(RNAi)* animals and controls (0.00311 vs 0.0297 control), $P = 0.595$, Mann–Whitney Rank Sum Test, two-sided, $N = 210$ cells for each group, three independent experiments. **c** no significant difference in allele bias for fluorescently tagged *idh-1* reporter alleles between *met-2(RNAi)* animals and controls (0.00337 vs 0.00356 control), $P = 0.730$, Mann–Whitney Rank Sum Test, two-sided, $N = 208$ cells for control, $N = 209$ cells for *met-2(RNAi)*, three independent experiments. **d** Intestine cells from *eef-1A.1(null)* animals had a significantly higher median intrinsic noise than controls (0.00944 compared to 0.00254 for controls); $P < 0.001$, Mann–Whitney Rank Sum Test, $N = 180$ cells for each group, three independent experiments. *eef-1A.1* reporter alleles were labeled with mTagBFP2 and mNeptune.

*lin-61(RNAi)* and *set-25(RNAi)* animals showed an extreme BAE phenotype, similar to controls (Fig. 4e, f). We conclude that *lin-65* and *arle-14* are also negative regulators of aRMAE, similar to *met-2*, whereas *lin-61* and *set-25* are positive regulators of aRMAE. Put differently, *met-2*, *lin-65*, and to a lesser extent *arle-14* promote a biallelic phenotype, whereas *set-25* and *lin-61* promote a monoallelic phenotype.

To determine if loss of LIN-61 or SET-25 activity would suppress the increased aRMAE observed in *met-2(null)* animals, we quantified intrinsic noise for *hsp-90* reporter alleles in *met-2(null);lin-61(RNAi)* and *met-2(null);set-25(RNAi)* animals. Both double mutants had a significant decrease in aRMAE (they showed a biallelic phenotype), demonstrating that the increase in monoallelic expression when MET-2 is genetically ablated is dependent on SET-25 and LIN-61, shown in Fig. 4g, h. This was also true for intronless reporter alleles (Supplemental Fig. 3a, b) and *eef1A.1* reporter alleles (Supplemental Fig. 3c, d). Figure 4i shows representative images of the *hsp-90* double mutants and *met-2(null)* control animals.

In our initial screen for negative regulators of aRMAE, *set-25* had a coefficient of determination slightly higher than controls (Table 1). Therefore, we screened all other genes on Table 1 that had a coefficient of determination above controls ($R^2 > 0.928$) for the ability to suppress the increased aRMAE phenotype in *met-2(null)* mutants (Supplemental Fig. 4). We identified one additional suppressor of aRMAE, *hpl-2*, a heterochromatin protein 1 homolog (See Supplemental Figs. 4 and 5). HPL-2/HP1 regulates chromatin, stress resistance, and aging in *C. elegans*[58]. Taken together, these results show that *set-25* and *met-2* function antagonistically to regulate aRMAE, and the mechanism includes HPL-2 and LIN-61, which likely act as accessories to SET-25 function. Figure 4j shows the genetic pathway supported by our experiments.

## MET-2 and SET-25 require catalytic SET domains to regulate MAE

MET-2 and SET-25 are both H3K9 HMT proteins that likely utilize a catalytic SET domain to transfer methyl groups onto H3K9. However, in the context of germline immortality[59], MET-2 also has an antagonistic relationship with SET-25 that reportedly does not require MET-2 HMT activity[45]. Thus, because MET-2 has biological activity independent of its HMT SET domain, SET-25 and MET-2 may or may not require SET domains for regulating aRMAE.

To test the hypothesis that the regulation of aRMAE requires the SET domains of MET-2 and SET-25, we used CRISPR/Cas9 to generate single residue changes in the catalytic domains of MET-2 and SET-25, and then measured aRMAE as described. We changed a critical cysteine residue to alanine, as was previously demonstrated to effectively nullify SET domain activity for *met-2*[45] (Fig. 5a–c). The resulting strains were *met-2(cat)*, which reprograms MET-2 for a C1237A residue change, and *set-25(cat)*, which reprograms SET-25 for a C645A residue change, making the putative corresponding SET domain mutation in SET-25. Additionally, we made a third strain with both *met-2(cat)* and *set-25(cat)* mutations. The *met-2(null)* and *met-2(cat)* mutants were phenotypically distinct from each other for fecundity and lifespan, and none of the mutants (*met-2(null/cat)* and *set-25(cat)*)

affected somatic RNAi initiation and inheritance (Supplemental Figs. 6 and 7).

We found the *met-2(cat)* animals phenocopied the *met-2(RNAi)* and *met-2(null)*, with all animals showing extreme aRMAE and higher intrinsic noise compared to controls (Fig. 5d). Next, we measured allele expression in *set-25(cat)* animals and found this mutation phenocopied the *set-25(RNAi)*, with lower intrinsic noise compared to controls, and a BAE phenotype, shown in Fig. 5e. Finally, we measured intrinsic noise in the *met-2(cat);set-25(cat)* double point mutation mutants, and found they had significantly lower intrinsic noise than both control animals and *met-2(cat)* animals, displaying a clear BAE phenotype (Fig. 5f). These results show that the antagonistic regulation of aRMAE by MET-2 and SET-25 requires the H3K9 HMT catalytic SET domains of both proteins to be intact.

## Maternal MET-2 and SET-25 act in the early embryo to control aRMAE

In *C. elegans*, the entire set of 20 intestine cells are descended from the single E-cell in the 8-cell embryo. Because we see persistent aRMAE throughout the entire intestine in some *met-2(null)* animals, it suggests that aRMAE can be initiated in the E-cell, propagated throughout mitotic divisions, and maintained into adulthood (e.g., Fig. 1a, d). Throughout this study, we observed that about 50% of *met-2(null)* animals had at least 50% or more of intestine cells in the same MAE state (~25% had almost or entirely monoallelic intestines), indicating that the silencing was most likely initiated in the E-cell of these animals. In the 8-cell embryo, there is very tight control of embryonic gene transcription[60]. We hypothesized that because of the early initiation of aRMAE, we (here), and others[61] observe, and the rapid pace of *C. elegans* development, maternal or paternal proteins are responsible for initiation of aRMAE in the E-cell.

To test our hypothesis, we performed two sets of reciprocal crosses. First, we crossed *met-2(cat);P$_{hsp-90}$::mEGFP* males with *met-2(wt);P$_{hsp-90}$::mCherry* hermaphrodites, and performed the reciprocal cross of *met-2(wt);P$_{hsp-90}$::mEGFP* males with *met-2(cat); P$_{hsp-90}$::mCherry* hermaphrodites (Fig. 6a, b). When *met-2(cat)* was crossed in from the male, all progeny showed BAE (Fig. 6c), which is expected in a mutant that is heterozygous recessive. However, when *met-2(cat)* was crossed in from the female, all progeny showed aRMAE (Fig. 6d), despite being heterozygous for a recessive *met-2* mutation. The increase in median intrinsic noise when animals receive *met-2(cat)* from the female germline was highly significant, shown in Fig. 6e, f.

Next, we performed a cross of *met-2(cat);set-25(cat);P$_{hsp-90}$::mEGFP* males with *met-2(cat);P$_{hsp-90}$::mCherry* hermaphrodites (Fig. 6g), and the reciprocal cross of *met-2(cat);P$_{hsp-90}$::mEGFP* males with *met-2(cat);set-25(cat);P$_{hsp-90}$::mCherry* hermaphrodites (Fig. 6h). The F1 animals homozygous for *met-2(cat)*, and heterozygous for *set-25(cat)* received from the paternal germline, showed the expected aRMAE (Fig. 6i). This is the Mendelian phenotype that would be expected for a recessive heterozygous mutation in a positive regulator of aRMAE, in a homozygous negative aRMAE regulator background. The F1 progeny

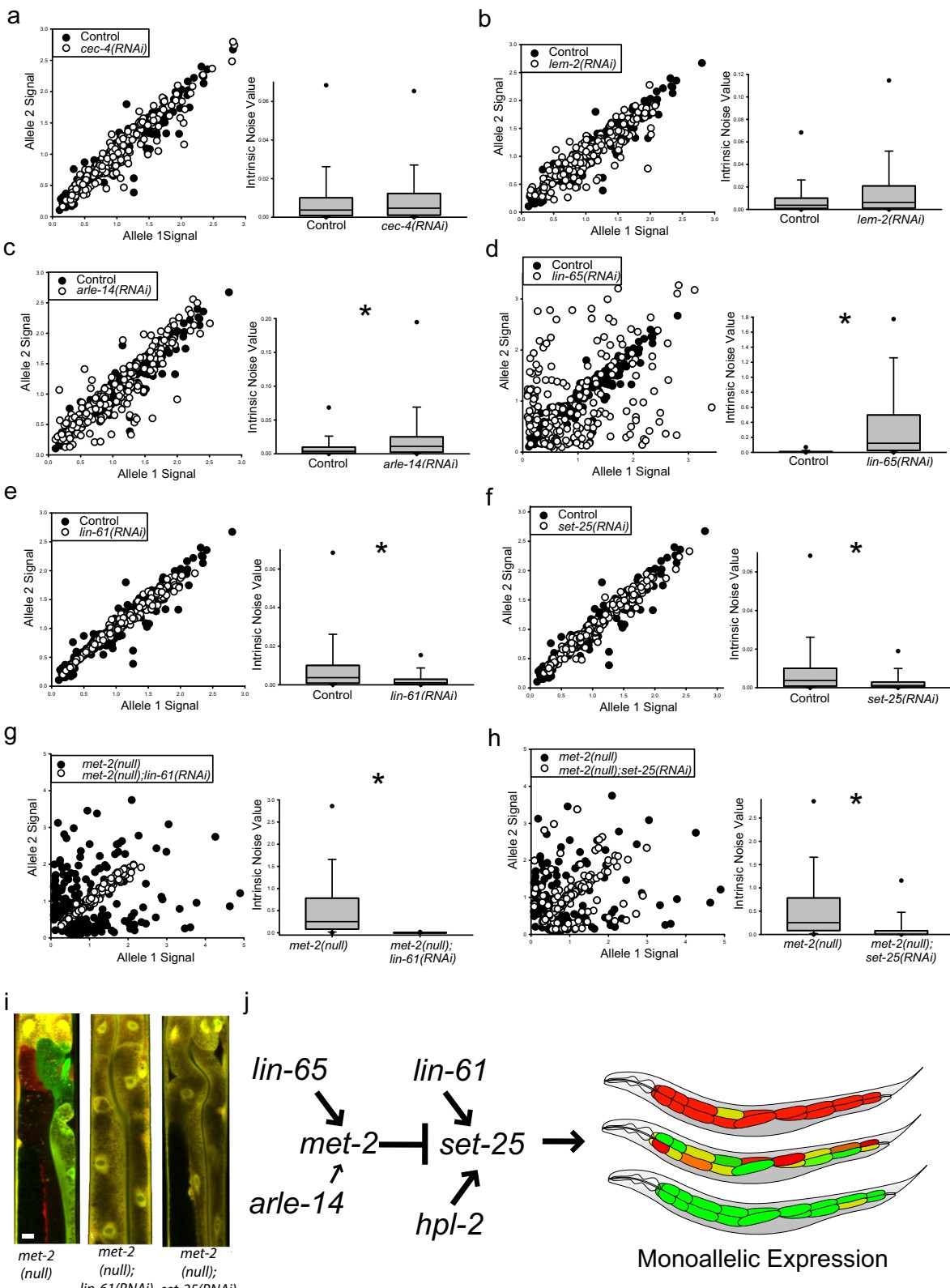

homozygous for *met-2(cat)* and heterozygous for *set-25(cat)* from the maternal germline showed unexpected BAE (Fig. 6j). This result is inconsistent with the expected Mendelian phenotype for a heterozygous recessive mutation in a positive regulator of aRMAE, in a negative aRMAE regulator background. The median intrinsic noise for F1 progeny heterozygous for *set-25(cat)* was significantly greater for animals that received the mutant *set-25* from the maternal germline,

compared to animals that received the mutant *set-25* from the paternal germline, shown in Fig. 6k, l. These results show that aRMAE is regulated in the early embryo by the catalytic SET domain activity of maternal MET-2 and SET-25. Consistent with random silencing in the early embryo, we also found the pattern of aRMAE in *met-2(cat)* animals was not heritable (Supplemental Fig. 8). A model based on our findings is shown in Fig. 7.

**Fig. 4 | A unique genetic pathway regulates monoallelic expression.** For (**a**–**h**), scatter plots (left panel) and boxplots (right panel) are all from intestine cells from animals with indicated mutations or RNAi. Top of boxplot is 75th percentile, bottom of box is 25th percentile, line is median, top and bottom error bars are 90th and 10th percentile, respectively, and dots are 95th and 5th percentile. **a** *cec-4(RNAi)* had no significant effect on aRMAE, (0.00467 vs control 0.00371); *P* > 0.05, Kruskal−Wallis One Way Analysis of Variance followed by Dunn's Method, *N* = 207 cells for control, *N* = 210 cells for *cec-4(RNAi)*, three independent experiments. **b** *lem-2(RNAi)* had no significant effect on aRMAE, (0.00624 vs control 0.00371); *P* > 0.05, Kruskal−Wallis One Way Analysis of Variance followed by Dunn's Method, *N* = 207 cells for control, *N* = 210 cells for *lem-2(RNAi)*, three independent experiments. **c** *arle-14* negatively regulates aRMAE, with a significantly higher median intrinsic noise of 0.0110 for *arle-14(RNAi)* compared to 0.00371 for control animals; *P* < 0.05, Kruskal−Wallis One Way Analysis of Variance on Ranks followed by Dunn's Method, *N* = 207 cells for control, *N* = 210 cells for *arle-14(RNAi)*, three independent experiments. **d** *lin-65* is a strong negative regulator of aRMAE, with a significantly higher intrinsic of 0.0110 for *lin-65(RNAi)* vs 0.00371 for controls, *P* < 0.05, Kruskal−Wallis One Way Analysis of Variance on Ranks followed by Dunn's Method, 207 cells for each group, three independent experiments. **e** *lin-61* is a strong positive regulator of aRMAE. *lin-61(RNAi)* animals had a significantly lower intrinsic noise of 0.00105 compared to 0.00371 for control animals; *P* < 0.05, Kruskal−Wallis One Way Analysis of Variance on Ranks followed by Dunn's Method, *N* = 207 cells for control, *N* = 210 cells for *lin-61(RNAi)*, three independent experiments. **f** similar to *lin-61*, *set-25* is also a strong positive regulator of aRMAE. *set-25(RNAi)* animals had a significantly lower intrinsic noise of 0.00109 compared to 0.00371 for control animals; *P* < 0.05, Kruskal−Wallis One Way Analysis of Variance on Ranks followed by Dunn's Method, *N* = 207 cells for control, *N* = 208 cells for *set-25(RNAi)*, three independent experiments. **g** The enhanced aRMAE phenotype in *met-2(null)* animals is dependent upon *lin-61*. The *met-2(null);lin-61(RNAi)* animals had a significantly lower intrinsic noise of 0.00160 compared to 0.254 for *met-2(null)* animals; *P* < 0.05 Kruskal−Wallis One Way Analysis of Variance on Ranks followed by Dunn's Method for multiple comparison, *N* = 199 for *met-2(null)*, *N* = 210 for *met-2(null);lin-61(RNAi)*, three independent experiments. **h** Similar to (**g**), the enhanced aRMAE phenotype in *met-2(null)* animals is dependent upon *set-25*. The *met-2(null);set-25(RNAi)* animals had a significantly lower intrinsic noise of 0.00237 compared to 0.254 for *met-2(null)* animals; *P* < 0.05 Kruskal−Wallis One Way Analysis of Variance on Ranks followed by Dunn's Method for multiple comparison, *N* = 199 cells for *met-2(null)*, *N* = 204 cells for *met-2(null);set-25(RNAi)*, three independent experiments. **i** Merged confocal microscope images of *met-2(null)* control, *met-2(null);lin-61(RNAi)*, and *met-2(null);set-25(RNAi)* animals with a 10 micrometer white scale bar inset in the bottom left of the control animal. **j** Summary of the genetic pathway controlling aRMAE from the results. Additional statistical and genetic comparisons are shown in Supporting Information Section 2; see Supplemental Fig. 3 for additional genetic tests.

## Discussion

### Our model for regulation of monoallelic expression

We report a working model for a developmental genetic pathway that controls aRMAE in the worm soma. It consists of two known H3K9 HMTs working antagonistically along with known associated proteins. Our data suggests that HPL-2 and LIN-61[43] enable maternal SET-25 to bind to histones associated with promoters of genes with aRMAE potential, and for intestine tissue, this can happen early in development. In parallel, the MAE-related silencing activities of HPL-2, LIN-61, and SET-25 are inhibited by LIN-65, MET-2, and to a lesser extent ARLE-14. LIN-65 translocates MET-2 to the nucleus[56], where it is helped by ARLE-14[56] to act as a transcriptional-silencing-repressor of SET-25, affecting the timing of initiation of SET-25 based silencing to generate different patterns of allele expression.

Prior work has shown that SET-25 regulates heterochromatin and silences the expression of genes and transposons via two different transposon control pathways[43]. The first is the MET-2-LIN-61-SET-25 heterochromatin control pathway in which SET-25 works cooperatively with LIN-61 and MET-2 to trimethylate histones in heterochromatin and at transposon loci[43]. In the second control pathway, SET-25 functions with NRDE-3 to add all three methyl groups to H3K9. Here, we found SET-25 and LIN-61 promote the silencing of alleles in the absence of MET-2 (Fig. 4). The function of LIN-61 in regulating MAE is not completely clear, but it may be generally assisting SET-25 to bind H3K9[62]. Similarly, we detected no effect of *nrde-3(RNAi)* on MAE in our initial screen or in subsequent experiments (see Table 1 and Supplemental Fig. 2). Thus, the current six-gene network performing MAE regulation in *C. elegans* overlaps with known genetic pathways regulating other silencing-related phenomenon, but seems unique in its utilization of genetic components.

If MET-2 uses its catalytic SET domain to inhibit SET-25 activity, it is most likely depositing H3K9 markings, based on our current understanding of MET-2 activities. SET-25 can methylate H3K9 independently of MET-2[42–44]. If SET-25 HMT activity is inhibited by MET-2 HMT activity, then there are two plausible scenarios for probabilistic regulation of non-heritable aRMAE. One scenario is that SET-25 cannot initiate methylation on H3K9me1. In this scenario, H3K9me1 put down by MET-2 leads to expression of an allele and inhibition of SET-25 HMT activity. However, if MET-2 marks histones with H3K9me2, or if MET-2 is inhibited from methylating H3K9 altogether, then SET-25 can add the third methyl to H3K9me2, or it could add three methyl groups to H3K9me0 to inhibit expression of an allele. In this scenario, SET-25 cannot act on H3K9me1 or it can do

so with low efficiency. This scenario explains an antagonistic relationship wherein both MET-2 and SET-25 require their SET domains for H3K9 methylation at aRMAE loci. This is similar to how MET-2 regulates methylation of germline genes in the soma[44].

In another scenario, SET-25 and MET-2 could be working with other yet unidentified proteins to regulate MAE via H3K9 methylation states. Alternatively, SET-25 and MET-2 could have additional SET-related activities that modify other residues, perhaps being part of a combinatorial code[14,15] or even new marks[63]. ChIP based studies in mammalian tissue found the H3K9me3 marking to be non-informative for ChIP based identification of MAE[14,15], consistent with a non H3K9me role for MET-2 and SET-25. Yet, in other biological contexts, H3K9 marks are the critical determinants of monoallelic expression[21]. We hypothesize that in prior genome-wide ChIP studies, the 5mC and H3K9me3 signals from the promoters were dwarfed by the H3K27 signals in the gene bodies[14,15]. In this scenario, the silencing would be licensed by the methylation of 5mC or H3K9 in promoter regions and then executed by the H3K27me3 marks in the gene bodies. Here, we found no effects of H3K27 HMTs in our screen.

Regulation of aRMAE is gene specific[12,14,15,21,48], so it is not surprising that not all of our reporter allele strains responded similarly to *met-2*. We and others have determined that *hsp-90*, *hsp-16.2*, *idh-1*, and *eef1A.1* all have aRMAE potential (this publication, and refs. 22,34). Yet, in this study, reporter alleles for *hsp-16.2* and *idh-1* did not respond to *met-2* perturbation, while *hsp-90* and *eef1A.1* alleles did. This is consistent with reports of aRMAE of a subset of genes being regulated by DNA methylation[12], or other epigenetic marks[19,21]. We have yet to identify the genetic pathways regulating *hsp-16.2* or *idh-1* (*idh-1* may only go monoallelic in tumors). However, the framework that we defined here can be used to do identify regulators of aRAME in worms and other model systems. Moreover, technologies such as Fiber-seq[64], which can be used to examine nucleosome occupancy changes at single-molecule resolution, can be used in conjunction with genetic studies to help define additional aRMAE control pathways.

### Outlook and translation

Here, we have worked toward an understanding of the play-by-play events regulating the initiation, propagation, and maintenance of monoallelic expression[8]. We discovered that two conserved H3K9 HMTs and their accessory proteins antagonistically control MAE in the early embryo. This developmental genetic pathway can explain how monoallelic expression is initiated and propagated throughout a tissue. The homologs of genes we identified here may regulate MAE in mammalian

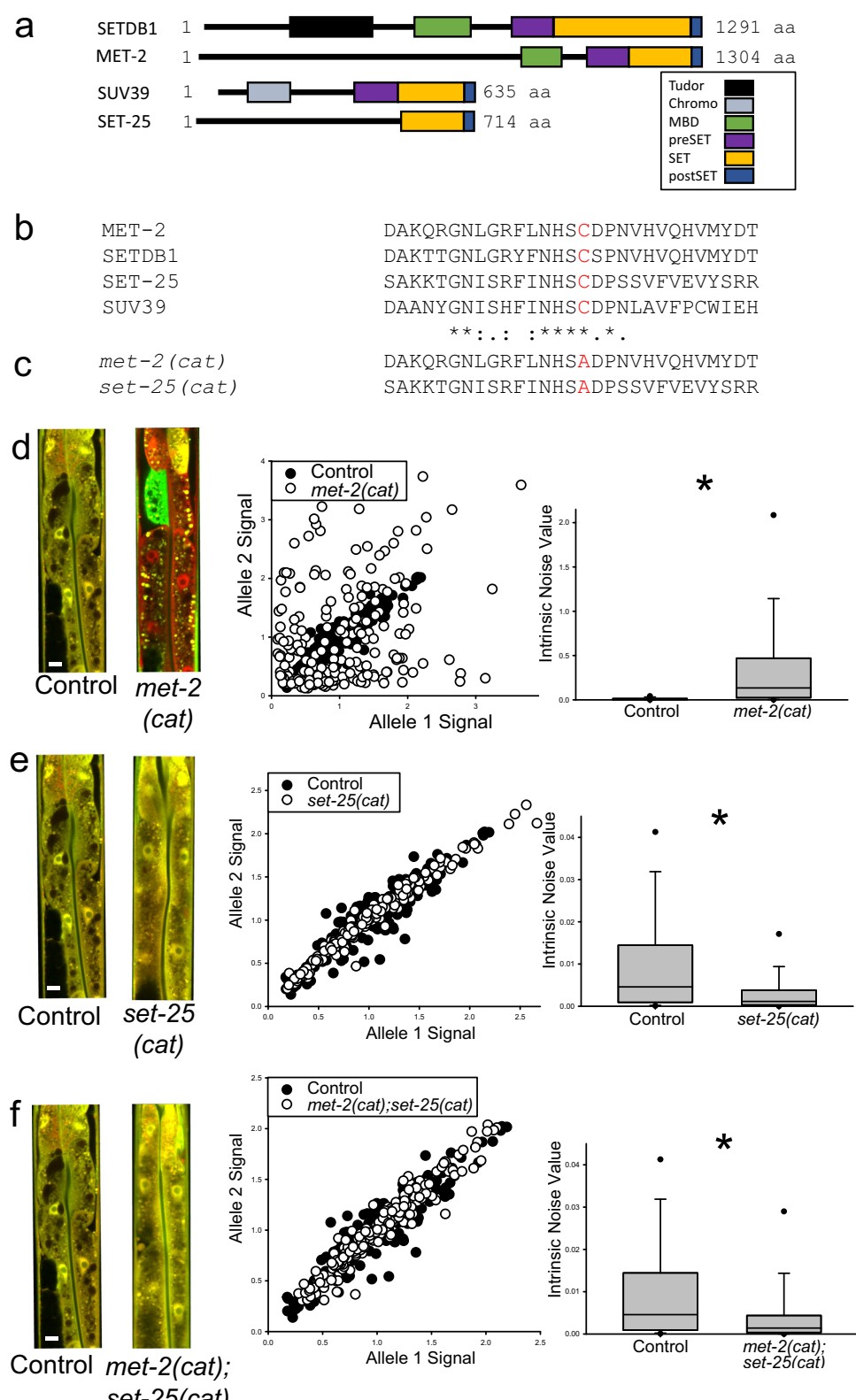

systems, where H3K9 methylation is known to control MAE-related silencing[21]. This model system can be used to identify additional MAE regulatory pathways and corresponding changes to chromatin.

Further investigations will enable us to understand the gene-specific mechanisms that control random autosomal monoallelic expression. Moreover, studies of how the silencing of individual alleles are naturally controlled are crucial to understand how to better

implement our own allele specific silencing and activating technologies to positively affect human health, such as the penetrance of inborn errors of immunity[19] or genetic diseases[18]. Finally, the assessment of silencing states to make treatment decisions is already here, as ovarian cancer patients with silenced *BRCA1/2* now have more treatment options[32]. Understanding the mechanisms that control aRMAE in both normal and disease tissue has the potential to provide avenues

**Fig. 5 | Cysteine to alanine mutation in the SET domain of SET-25 or MET-2 causes loss of monoallelic expression regulation.** For each row, (**d**–**f**), left panel "torso" images of control and experimental worms with a 10 micrometer white scale bar inset in the bottom left of each control animal; middle panel scatter plots of control and experimental group intestine cells plotted by allele expression values from all three independent experiments; right panel) boxplots of intrinsic noise values from cells from control and experimental worms. Top of boxplot is 75th percentile, bottom of box is 25th percentile, line is median, top and bottom error bars are 90th and 10th percentile, respectively, and dots are 95th and 5th percentile. **a** Graphical alignment of human SETDB1 and SUV39 with worm homologs. Predicted domains are highlighted by different colored boxes. **b** Text alignment of MET-2, SETDB1, SET-25, and SUV39 with conserved cysteine residue in red. **c** Text alignment of the SET domain of *met-2(cat)* and *set-25(cat)* mutants. These mutations change a conserved cysteine to alanine. **d** *met-2(cat)* phenocopies the null mutant (Fig. 2a), with significantly higher intrinsic noise than control animals;

0.134 for *met-2(cat)* vs. 0.00459 in controls, $P < 0.05$, Kruskal–Wallis One Way Analysis of Variance on Ranks followed by Dunn's Method for Multiple Comparison, $N = 209$ cells for control, $N = 210$ cells for *met-2(cat)*, three independent experiments. **e** *set-25(cat)* phenocopies the *set-25* RNAi (Fig. 4f), with significantly lower intrinsic noise than controls; 0.00114 in *set-25(cat)* vs 0.00459 for control, $P < 0.05$, Kruskal–Wallis One Way Analysis of Variance on Ranks followed by Dunn's Method for Multiple Comparison, $N = 209$ cells for control, $N = 210$ cells for *set-25(cat)*, three independent experiments. **f** *met-2(cat);set-25(cat)* double SET- domain mutant animals have significantly lower intrinsic noise than *met-2(cat)* animals and *hsp-90* control animals; 0.00140 for *met-2(cat);set-25(cat)* vs. 0.00459 for control or 0.134 for met-2(cat), $P < 0.05$ for both comparisons, Kruskal–Wallis One Way Analysis of Variance on Ranks followed by Dunn's Method for Multiple Comparison, $N = 209$ cells for control, $N = 210$ cells for *met-2(cat)*, 210 cells for *met-2(cat);set-25(cat)*, three independent experiments.

for development of new diagnostics, and to treat recalcitrant diseases like cancer.

## Methods

### Strain creation
We made all our own strains. We used alt-R CRISPR/Cas9 (IDT, Newark, NJ) to generate endogenous gene tags and point mutations[13]. For making single point mutations in the *met-2* and *set-25* catalytic domains, and for making the *met-2(null)*, we used ssODNs as repair templates (IDT, Coralville, IA). For tagging endogenous *idh-1*, we used PCR products as repair templates. CRISPR target and repair template sequences are listed in Supplemental Table 2. Primers are listed in a Supporting Information Excel File. CRISPR point mutations were made directly into the heterozygotic GFP/mCherry reporter allele expressing strains. Genomic DNA was harvested from each CRISPR edit and a region spanning the repair template was amplified by PCR and sequenced (Genewiz, Burlington, MA, Supplementary Fig. 9). The resulting strain names and genomic insertion designations are shown in Supplemental Table 1.

### Animal husbandry and RNAi
We maintained all strains in 10 cm petri dishes on NGM seeded with OP50 *E. coli* in an incubator at 20°. All experiments were performed at 20°, with the exceptions of a heat shock experiment series and male generating procedures noted below. RNAi experiments used NGM supplemented with 50 µg/ml carbenicillin, 10 µg/µl tetracycline, and 1 mM IPTG. All strains used in this study are listed in Supplemental Table 1. A table of crosses can be found in Supplemental Table 3. To generate heterozygous GFP/mCherry expressing strains, we generated GFP expressing males by subjecting 20 L4s to a 30° heat shock for 5–6 h. For all strains, the GFP allele was introduced through the male germline. We screened for heterozygous animals that express both GFP and mCherry on a fluorescence stereoscope. We maintained heterozygotes by picking them away from homozygous animals and onto fresh, OP50-seeded NGM growth plates each generation. We performed experiments on heterozygous animals that were at least five generations beyond the initial cross to avoid paternal allele expression bias. To synchronize animals for experiments, we conducted two hour egg lays onto 10 cm NGM or RNAi plates (10 heterozygous animals per plate). For experiments with heat shock to quantify *hsp-16.2* reporter expression, we performed a one-hour heat shock at 35° on one day old adult animals by placing animals on their NGM growth plates into a 35° incubator for one hour and then returned them back to the 20° incubator until imaging the next day (24 h post-heat shock). Additional details available in Brenner, 1974[65].

RNAi stocks were pulled from stock media containing ampicillin, tetracycline, and chloramphenicol. All RNAi clones were sequenced prior to seeding NGM plates. For RNAi experiments, we used only clones that had been reported to have effects (for example, *cec-4* and *lem-2* clones were used in Towbin et al., 2012). In addition, we used an mCherry-expressing worm strain and anti-mCherry RNAi as an experimental control. For experiments, day 2, adult worms heterozygous for reporter alleles were placed on RNAi food or EV to lay eggs for two hours before the parents were removed. The process was repeated on day 4 for a total of two generations on RNAi food. On day eight from the start of the experiment, synchronous day 2 adults were harvested for microscopy.

### Progeny production per individual
To quantify differences in the fertility of the self-fertile *C. elegans* hermaphrodites, we picked L4 animals onto single plates and allowed individual animals to lay eggs for the first four days of adulthood at 20°. P0 worms were moved to new plates each day. We quantified progeny per individual per day by counting the number of hatched progeny at the L4/YA age, two days after moving the hermaphrodite. Progeny for all worms were measured approximately 20 generations after the worms were created.

### Lifespan measurements
To quantify genetic effects on aging, we quantified lifespan using the semi-automatic WormBot system[66]. Briefly, L4 animals are placed in groups of approximately 20–30 onto OP50-seeded NGM media supplemented with 50 µM FUDR to inhibit progeny production in 12-well plates with clear plastic bottoms and lids. The WormBot system captures images/movies of each well that we then manually review for the cessation of movement/death.

### RNAi kinetics in mutant strains
To determine if the somatic RNAi machinery was functional and if RNAi heritability was altered in *met-2* and *set-25* mutants, strains homozygous for $P_{hsp-90}::mCherry$ reporters were grown on OP-50 food for at least five generations before conducting RNAi. For egg lays, 10 gravid adults were placed on RNAi food or OP50 for two hours to lay eggs before removing the P0s. On experiment days 4 and 8, egg lays were conducted on RNAi food. On experiment days 12 and 16, egg lays were conducted on OP50 food. After each egg lay, the remaining worms from each plate were harvested for imaging. To image, 15–20 worms from each plate were anesthetized on agarose slides and imaged on a Zeiss LSM780 confocal microscope with a 10× air objective.

### Heritability experiments
To determine if patterns of monoallelic expression were heritable, we used a Leica M165F Fluorescent stereoscope with a 500 nm long-pass emission filter to identify day 2 adult *met-2(wam406)* individuals that were heterozygous for fluorescent red and green *hsp-90* reporter alleles. We picked individuals with either variegated or monoallelic expression patterns in their intestine onto individual plates. We let

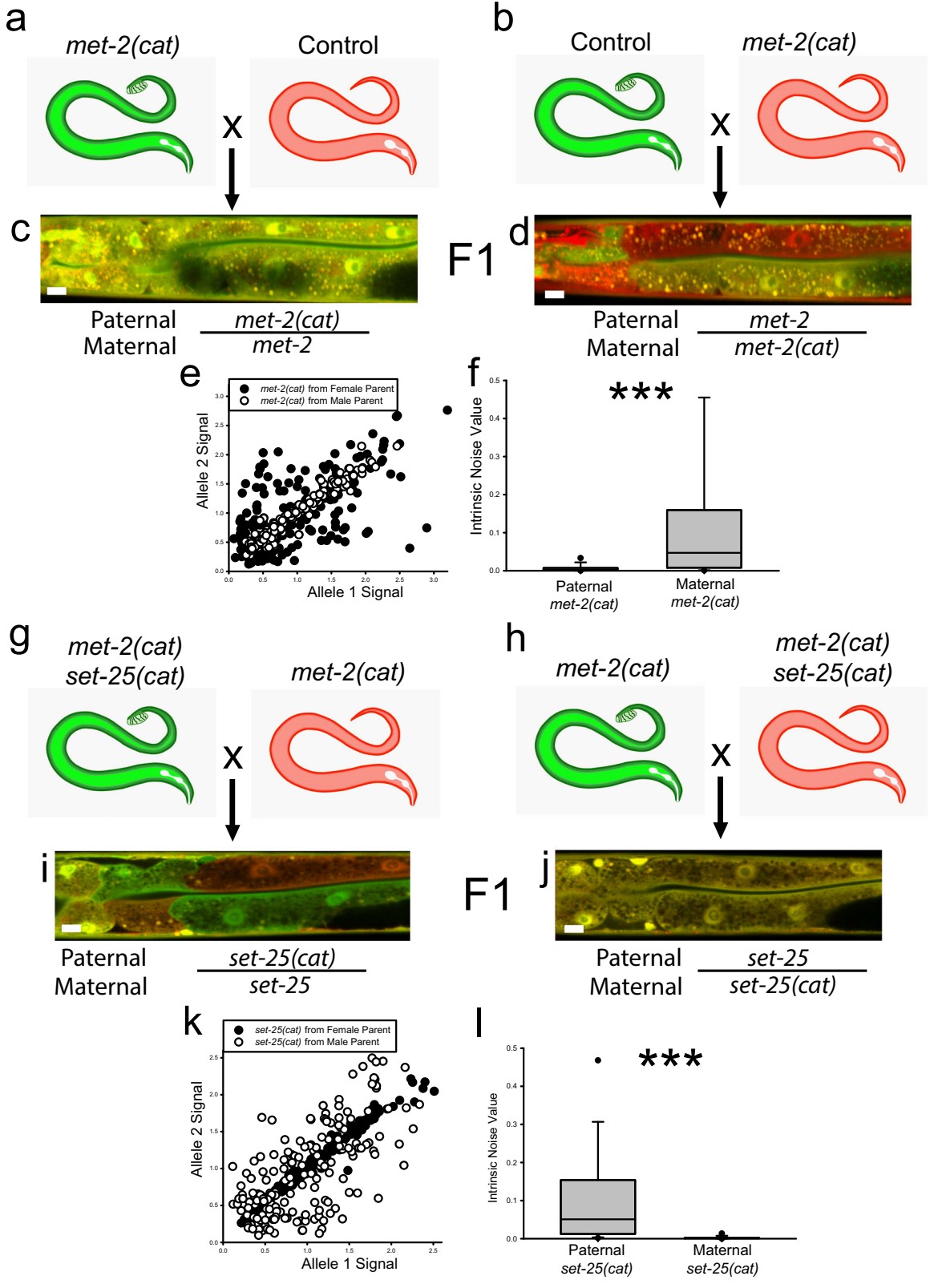

each individual animal lay eggs on a 6 cm NGM plate for four hours, we then mounted each individual animal on an agarose pad to record an image by confocal microscopy as we describe below. Four days later, we loaded progeny from each individual into a microfluidic chip for imaging as we describe below. We imaged animals that were detectably heterozygous for reporter alleles and manually compared patterns of expression to parents. There is an obvious lack of correlation

between parents and progeny, shown in Supplementary Fig. 7, which is representative of the individuals we imaged over three independent experiments.

**Reciprocal crosses/maternal or paternal germline contribution**
To determine if the H3K9 HMT catalytic SET domain of MET-2 from maternal and paternal germ cells was acting to regulate monoallelic

**Fig. 6 | Maternal MET-2 and SET-25 act in the early embryo to regulate mono-allelic expression. a, b** show reciprocal crosses generating F1 hybrids that are heterozygous for recessive *met-2(cat)*, with the mutation coming from either the paternal (**a**) or maternal (**b**) germ line. *met-2(cat)* from the paternal germline leads to BAE in F1 heterozygotes (**c**), whereas *met-2cat* from the maternal germline leads to aRMAE (**d**); a 10 micrometer white scale bar is inset in the bottom left of each animal. **e** Scatterplots of intestine cells from the animals in (**c** and **d**). **f** Boxplots of intrinsic noise values from each cross. Top of boxplot is 75th percentile, bottom of box is 25th percentile, line is median, top and bottom error bars are 90th and 10th percentile, respectively, and dots are 95th and 5th percentile. Heterozygotes that got the *met-2(cat)* allele from the maternal germline had significantly higher intrinsic noise than animals that got the same allele from the paternal germline; 0.0469 median intrinsic noise for maternal *met-2* vs 0.00193 for paternal *met-2*, *P* < 0.001, Mann–Whitney Rank Sum Test, two-sided, *N* = 180 cells per group, three independent experiments. Despite being heterozygous for *met-2(cat)*, animals got the mutant allele via the maternal germline had similar phenotypes to *met-2(RNAi)* and *met-2(null)*, shown in Figs. 1c and 2a. **g, h** show reciprocal crosses generating F1

hybrids that are homozygous for *met-2cat* and heterozygous for *set-2(cat)*, with the *set-25(cat)* mutation coming from either the paternal (**g**) or maternal (**h**) germline; a 10 micrometer white scale bar is inset in the bottom left of each animal. Homozygous *met-2(cat)* results in aRMAE, however, when the *set-25(cat)* mutation is passed through the maternal, but not the paternal germline, the result is a BAE phenotype in *met-2(cat);set-25(cat)* animals (**i, j**). **k** shows scatterplots of cells from F1 animals from each reciprocal cross shown. **l** shows boxplots of intrinsic noise of cells from F1 animals from each reciprocal cross shown. Top of boxplot is 75th percentile, bottom of box is 25th percentile, line is median, top and bottom error bars are 90th and 10th percentile, respectively, and dots are 95th and 5th percentile. The F1 *met-2(cat)* homozygous, *set-25(cat)* heterozygous animals that got the *set-25(cat)* allele from the maternal germline had the same decreased intrinsic noise as homozygous *set-25* mutants and RNAi (shown in Figs. 4f and 5e), with a significantly lower intrinsic noise than animals that got the mutant *set-25(cat)* allele from the paternal germline; 0.00109 median intrinsic noise for maternal mutant *set-25* vs. 0.0508 for paternal mutant *set-25*, *P* < 0.001, Mann–Whitney Rank Sum Test, two-sided, *N* = 180 cells per group, three independent experiments.

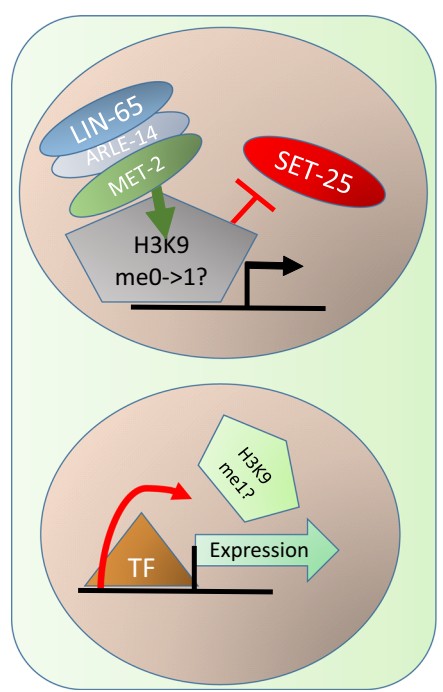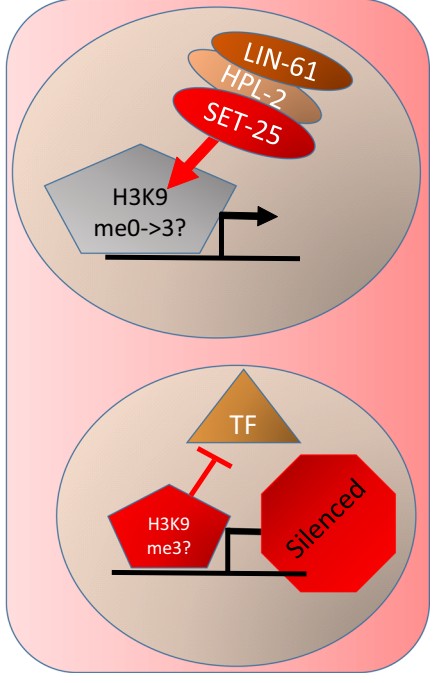

**Fig. 7 | Our model of monoallelic regulation.** Left panel Biallelic expression. MET-2, with the help of ARLE-14 and LIN-65, bind to a gene with monoallelic expression potential, allowing MET-2 to methylate H3K9, and block SET-25 binding (Upper Left). This results in transcription factor (TF) binding and biallelic expression (Left

lower). Right panel Monoallelic expression. SET-25 is recruited to a gene with monoallelic potential before MET-2, or displacing MET-2. SET-25 can then tri-methylate H3K9 (Upper right). This results in transcription factor blocking, and the silencing of an allele. The end result is monoallelic expression (Lower right).

expression in the early embryo, specifically the E-cell, we conducted crosses with *met-2(cat);P_{hsp-90}::mEGFP* homozygous males with *P_{hsp-90}::mCherry* homozygous hermaphrodites. We performed reciprocal crosses with *P_{hsp-90}::mEGFP* homozygous males crossed to *met-2(cat);P_{hsp-90}::mCherry* homozygous hermaphrodites.

To determine the contribution of the SET-25 catalytic domain via maternal and paternal germline, we conducted crosses with *met-2(cat);P_{hsp-90}::mEGFP* homozygous males with *met-2(cat);set-25(cat);P_{hsp-90}::mCherry* homozygous hermaphrodites. We performed reciprocal crosses with *met-2(cat);set-25(cat);P_{hsp-90}::mEGFP* homozygous males crossed to *met-2(cat);P_{hsp-90}::mCherry* homozygous hermaphrodites.

**Microscopy**

We washed day 2 adult animals (second day of adulthood at 20°) into S-basal media with tricaine/tetramisole[37], and loaded animals

into 80-lane microfluidic devices[40]. These devices immobilize worms in 80 separate lanes in a relatively restricted position, making presentation of the animals to the objective more uniform than using traditional agarose based slides. We imaged only those animals that randomly immobilized with their left side facing the cover slip, to which the fluidic device was bonded, which put intestine cells in rings I through IV closest to the microscope objective. Doing this avoids quantification error due to loss of signal with depth of tissue (i.e., imaging intestine through the germline when animals orient on their right sides).

We imaged animals as we previously described[37]. Briefly, we used a 40 × 1.2 NA water objective on a Zeiss LSM780 confocal microscope. We excited the sample with 488 and 561 nm lasers and collected light from 490 to 550 nm for mEGFP signal, and from 580 to 640 nm for mCherry signal. For mTagBFP2 and mNeptune, we excited samples with 405 and 561 nm lasers and collected emission

from 410 to 480 nm and 570 to 650 nm, respectively. We also collected transmitted light signal for Nomarski DIC images to aid in cell identification as needed. We focused on the same field of view for each animal- starting from the posterior of the pharynx to the first half of cells in intestinal ring IV. We collected images of the entire z depth of each animal, from one side to the other, using two-micrometer step size and a two-micrometer optical slice as we have previously described[37]. Imaging settings were held constant between control and experimental groups.

## Image cytometry
Our image cytometry consists of manual cell identification and annotation, with a semiautomatic quantification step. We first determined the orientation of the animals in images and then identified individual intestine or muscle cells. We then measured signal within an equatorial slice of the cell's nucleus, as a proxy for the whole cell. Nuclear signal of freely diffusing monomeric fluorescent protein is nearly perfectly correlated with the cytoplasmic contents[37]. We used ImageJ software as well as custom-built Nuclear Quantification Support Plugin called C. Entmoot (Alexander Seewald, Seewald Solutions, Inc., Vienna, Austria) for nucleus segmentation and signal quantification[38]. Image cytometry here was performed essentially in the same manner as in Mendenhall et al.[37], which also shows a schematic cartoon of how we optically slice the nucleus with the confocal microscope.

## Data processing and noise calculations
Here, we measured intrinsic noise by measuring the expression level of differently colored reporter alleles in two-day old adult animals that are in a steady-state of gene expression[38]. Intrinsic noise is essentially the quantitative measure of relative deviation from the 1:1 ratio; data points having a 1:1 ratio fall on a 45° diagonal trend line. Intrinsic noise measures how deviant a pair of reporter alleles is from the average ratio among groups of cells, thus quantifying how probable it is to observe biased or monoallelic expression for a given gene (pair of alleles) in a given population of cells (e.g., muscle cells or intestine cells). The assumptions of our intrinsic noise model are the same as the assumptions in ref. 37. We sometimes used 8-bit or 16-bit file settings during data collection, though this difference was obviated after normalization. We normalized expression level data for each allele to per-experiment means as in previous investigations[67–69]. We calculated intrinsic noise as detailed in refs. 67–69. Specifically, the formula for calculating intrinsic noise is:

$$\text{Intrinsic noise} = ((x - y)^2)/(2\langle x\rangle\langle y\rangle)$$

where x and y are each cell's allele expression values and <x> and <y> are the average value for each allele. X,Y expression data and calculated noise values for each figure are available in Source Data Files, in Excel format.

## Statistics and reproducibility
We used SigmaPlot 12.5 (Systat Software, Inc., San Jose) for all plotting and statistical analyses. All data was non-normally distributed, even after attempting log or natural log transformations, thereby requiring nonparametric statistics for analysis. For experiments with multiple groups analyzing alleles in intestine cells or different promoters in intestine cells, we ran ANOVA on Ranks followed by Dunn's pairwise comparisons, unless otherwise noted. For all other experiments with only two groups, we ran a nonparametric Mann–Whitney U test for each distinct set of experiments. Details of each test are shown in Supplementary Information.

No statistical method was used to predetermine sample size. Sample size was empirical, based on Sands et al. 2021. No data were excluded from the analyses. The experiments were not randomized. The Investigators were not blinded to allocation during experiments and outcome assessment.

## Reporting summary
Further information on research design is available in the Nature Portfolio Reporting Summary linked to this article.

## Data availability
Source data are provided with this paper. Data are found in main text, Supporting Information Document and in Source Data Excel Files. Any additional relevant data are available from the corresponding author upon reasonable request. Source data are provided with this paper.

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

## Acknowledgements

Funding was provided by NCI R01CA219460 to A.M., the Nathan Shock Center for Excellence in the Basic Biology of Aging Invertebrate Healthspan and Longevity Core P30AG013280, a Pilot Grant to A.M. from the University of Washington EDGE Center of the National Institutes of Health funded by NIEHS P30ES007033, NCI R01CA210916 to J.O., and NIA R01AG063971 to J.O.

## Author contributions

A.M. and B.S. designed the study. B.S., A.M. and S.Y. performed experiments. B.S. and A.M. analyzed the data. B.S. and A.M. wrote the initial manuscript. B.S., S.Y., J.O. and A.M. revised the manuscript. A.M. and J.O. provided funding through grants.

## Competing interests

The authors declare no competing interests.
