## [Peer Review File · Nature Communications]

Maternal histone methyltransferases antagonistically regulate autosomal random monoallelic expression (aRMAE) in *C. elegans*

Corresponding Author: Dr Alexander Mendenhall

Version 0:

Reviewer comments:

Reviewer #1

(Remarks to the Author)

The article entitled "Maternal histone methyltransferases antagonistically regulate monoallelic expression in *C. elegans*" aims to reveal genetic pathways regulating the monoallelic expression (MAE) in the model species *C. elegans* during early development. For that purpose, the authors used a validated system of two distinctly colored fluorescent alleles. They used more classical molecular techniques such as RNAi or CRISPR-Cas 9 to identify genes regulating monoallelic expression in the digestive tract of this worm. The article is written in a sequential way with each conclusion justifying the following step. The main outcome of this research is the discovery of an antagonistic mechanism of two histone methyltransferases (*met-2* and *sep-25*) and associated proteins promoting MAE (*sep-25*) or BAE (*met-2*). This is quite original and can be a new benchmark to better understand MAE regulation mechanisms in *C. elegans* but also in other organisms including human. Conclusions are solid and supported by several experiments using different knockout and knockdown animals. The experimental design is effective and well presented. The methodologies are well chosen and extensively described in the M&M section. The figures are also well chosen and are informative by themselves. Beside this main conclusion, the authors also showed that this mechanism is not chromatin context and sequence dependent. However, it is not generalized as they could not reproduce the data with all the other tested genes, indicating that the regulation mechanism is clearly gene-dependent. They also reported that the SET domain is requested to regulate MAE. A second layer of conclusions are more related to adaptive and evolutionary outcomes as it emphasizes the role of MAE in epigenetic gambling. They supported this conclusion showing inter-individual variation in the production of progeny. They also showed and discussed the antagonistic control of lifespan by the two histone methyltransferases. This part of the manuscript is a bit more difficult to understand and I would suggest the authors to better explain the relevance of these conclusions. Lastly, they showed no heritability in MAE and the role of parental proteins in the initiation of MAE in early embryos. For that, they used a well-designed reciprocal crosses experiment.

Overall, this research tells a very interesting and original story about important epigenetic mechanisms in a common model species. The authors put together a huge amount of data and organized it in a comprehensive and logical way. The arguments are well discussed regarding the recent literature. I would therefore recommend the article for publication. However, I have few suggestions to improve the manuscript.

- the linear organization of the experiments and conclusions is relevant. However, it might dilute the main messages the authors would like to emphasize as there are a succession of small conclusions. I would suggest to split the conclusions in three parts: the mechanisms regulating MAE ; the epigenetic gambling and lifespan control ; the heritability.
- they reported a contradictory effect of *met-2* regarding what is known for transposons and heterochromatin but I would expect a development about this observation.
- the same authors reported in 2021 (Nat. Com. 12(1), 6527) the importance of introns in controlling stochastic allele expression bias. I would expect a few words here related to this mechanism.
- they introduced some concepts or issues regarding penetrance, human diseases or longevity but the discussion is quite limited regarding these points. For example, the final conclusion refers to application in human health but it's not really supported by the manuscript. Consequently, the perspectives and applications of the results are not really clear and could have been better explained.

Reviewer #2

(Remarks to the Author)

Reviewer #3

(Remarks to the Author)

Review:

The study by Sands et al. describes a new role for the histone methyltransferases MET-2 and SET-25 in regulating monoallelic expression (MAE) in *C. elegans*. They measure allele-specific expression using strains carrying two fluorescent protein transgenes inserted at the same locus and kept at a heterozygous state, which allows the quantification of the protein levels generated by each allele. By performing a candidate screening of factors belonging to chromatin and small RNA pathways, they found that loss of MET-2 results in mosaic expression of the transgene compared to control, indicating a shift towards monoallelic expression for two out of five promoters analyzed. They also found that the knockdown of the cofactor of MET-2, lin-65, also causes monoallelic expression of the transgenes. In contrast, the depletion of lin-61 and set-25 suppresses the monoallelic expression resulting from the absence of met-2. They next introduced mutations within the catalytic domains of MET-2 and SET-25 to show that their catalytic function is required for their effect on regulating monoallelic expression and that this regulation takes place during early developmental stages by maternally provided MET-2 and SET-25 proteins.

The observation that MET-2 and SET-25 have a role in regulating monoallelic expression is interesting and novel. Moreover, the impact of MET-2 and SET-25 at early developmental stages sheds light on how epigenetic mechanisms can regulate the establishment of monoallelic expression during development. However, further support should be provided since the evidence for MET-2 and SET-25 regulation of monoallelic expression comes mainly from analyzing transgenes driven by one promoter. In addition, allele-specific expression is inferred based on protein levels, and actual measurement of the transcriptional levels is not provided.

Specific comments to be addressed:

- 1- The authors have shown the effect of MET-2 on MAE for a single-copy transgene driven by the promoter of hsp-90. It would be important to observe similar effects on the endogenous hsp-90 gene. The authors have already generated such strain in Sands et al. 2021. Thus, they can study whether mutations in met-2 and/or set-25 affect the MAE of the endogenous hsp-90 gene. Moreover, according to Sands et al. 2021, introns prevent MAE on the hsp-90 gene and other transgenes. Thus, it would be interesting to test whether intron-less transgenes combined with met-2 mutation show increased MAE.
- 2- The authors showed that eef-1A.1 transgenic reporter also shows some degree of MAE in the met-2 mutant. However, they have not tested the effects of lin-65 RNAi on MAE nor the impact of set-25 and lin-61 in suppressing MAE in met-2 mutant. Testing these effects in eef-1A.1 transgenic reporter would reinforce their conclusion on at least two transgenes (with different promoters).
- 3- In the introduction, the authors state that: "met-2 mutant animals showed a pattern of monoallelic expression that was persistent throughout adulthood". The authors have only measured MAE in one developmental time point (adulthood). Similarly, in the introduction, the authors state, "Reciprocal cross experiments showed that maternal MET-2 and SET-25 act in the E-cell of the 8-cell embryo to regulate MAE". This may be the case for some animals with fully monoallelic tissues. However, this is not always the case since there are animals with mosaic tissues, indicating that monoallelic expression was determined at later developmental stages. The proportion of these scenarios (mosaic vs monoallelic) should be quantified.
- 4- The authors infer the transcriptional regulation of MAE by MET-2 from the quantification of protein expression in the hsp-90 fluorescent reporter. However, no mRNAs or transcription ratio quantification of the reporter Cherry and GFP has been shown. Using smFISH will give the resolution to test the effect of met-2 mutation in MAE at the transcriptional level.
- 5- The authors also aimed to test if MAE can be inherited. For this, they look at the allele-specific expression pattern in the progeny of met-2 mutant worms that showed mosaic transgene expression. They conclude that there is no inheritance of MAE because the progeny showed different allele-specific expression patterns than the parent, i.e., the parent had a mosaic intestine. In contrast, some progenies have a mostly monoallelic intestine (even though this was not quantified). The mosaic nature of monoallelic expression indicates that it is established randomly during development. Thus, it is unclear why the authors hypothesize that inheritance of MAE would imply inheritance of the pattern. To test if met-2 mutation could lead to epigenetic inheritance of monoallelic expression, the authors could cross out the met-2 mutation and assess if the resulting progeny still displays monoallelic expression.

Minor comments:

1. The authors can shorten the introduction to improve the manuscript's readability.
2. The authors can combine some figure panels to facilitate the follow-up of the paper (and reduce the number of figures)
3. The authors can perhaps substitute the name of the catalytic mutants with an acronym that facilitates the understanding of the catalytic mutation. For example, met-2(wam406) by met-2 CD.

Reviewer #5

(Remarks to the Author)

The authors designed an elegant screen for regulators of monoallelic gene expression (MAE) in *C. elegans* and identified histone methyltransferases that positively or negatively impact this process. The work is original because it is not sequenced

based and instead relies on in vivo expression of fluorescent reporters in a developmental context. This is important because single cell analysis that probes a single time point is unable to distinguish between dynamic monoallelic expression resulting from transcriptional bursting, and mitotically stable monoallelic expression. A limit of this approach may be that it relies on artificially constructed reporter alleles. Also, no evidence is provided for the role of MAE in animal physiology, although speculations are made.

Overall the methodology is sound, especially regarding reporter gene expression. However I have some reservations about the interpretation of the genetic data. In particular, stronger evidence for suppression of met-2 by set-25 is required. Main points are detailed below.

1/ figure 4; rescue of met-2 phenotype by set-25 catalytic dead allele

I see no evidence of rescue at day 3, contrary to what is stated in the legend (panel c). If anything the double set-25; met-2 is less fertile than the met-2 single.

In panel (e) total fecundity in double mutants is less variable than in single met-2, but the median looks the same.

2/ Given that the met-2(wam007) allele is more penetrant than the 406 allele, why did the authors not also test set-25; met-2(wam007) allele?

3/ could the authors show stats for significance on each panel rather than the legend ?

4/ for the catalytically dead alleles, did the author perform WB analysis or IF to show expression of full-length protein and that catalytic activity is indeed knocked down? Although it is predicted that these point mutations only affect catalytic activity, this should be confirmed

5/ Figure 4: efficacy of RNAi for genes that have no effect on DAE (cec-4, lem-2, lin-61, arle-1..) should be measured by RT-qPCR

6) lifespan assays do not really contribute to an understanding of how met-2 and set-25 contribute to MAE

In general, evidence linking germline immortality or longevity to somatic monoallelic expression is very weak. So although MET-2 and SET-25 are clearly involved in MAE, how MAE impacts animal physiology and the mechanism involved remains speculative.

Additional points

1/ In figure 3 it would help the reader to know the identity of the reporter alleles in each of the panels a-d. This is only explicitly stated for panel D in the legend (mTagBFP2 or mNeptune expression). The color of each fluorochrome should also be given: eg scarlet/violet

2/ The authors suggest that increased interindividual variation in progeny production of met-2 mutants (Figure 6f) may reflect monoallelic expression as a form of « epigenetic gambling » with some individuals having abundant progeny and others very few. While this may well be an explanation, inactivation of chromatin factors in *C. elegans* often results in reduced brood sizes, with decreases that can vary greatly between individuals

Version 1:

Reviewer comments:

Reviewer #1

(Remarks to the Author)

This revised manuscript presents now a compelling study of autosomal random monoallelic expression (aRMAE) in *C. elegans*, focusing on the antagonistic roles of the H3K9 histone methyltransferases MET-2 and SET-25. The authors have convincingly demonstrated that maternal MET-2 activity limits SET-25-driven silencing in early embryogenesis, thereby establishing persistent monoallelic expression patterns in the intestine. Compared to the initial version, the authors have responded accurately to the reviewers' comments. Moreover, they have added new data, with inclusion of intronless hsp-90 reporters and eef-1A.1 reporter. The overall conclusions are now more solid. The text has been substantially improved, mainly throughout a more focused and tightened discussion. Speculative sections on lifespan, immortality and epigenetic gambling have been deleted or diminished, and the text now centers on the mechanistic antagonism between MET-2 and SET-25 and its developmental timing. The authors have also changed some wording and concepts to refer to very recent literature. They are now using the term aRMAE instead of the more general MAE, which soundly refers to Stewart et al., 2025. The new figure 7 graphically summarizes their model of monoallelic regulation and is good visual synthesis. In conclusion, the revised version represents a significant improvement over the original. It delivers a clear mechanistic insight into how opposing histone methyltransferases establish and maintain aRMAE in *C. elegans*. The authors' rebuttal and the additional data address the reviewers' concerns and significantly increase the robustness of the conclusions. The manuscript, in its current form, is suitable for publication.

Reviewer #2

(Remarks to the Author)

We thank the authors for adequately addressing most of our concerns. We still believe that one limitation of the study is that it does not directly assess transcription as a read-out of monoallelic stochastic expression. Even though we acknowledge that methodological limitations prevent the authors from addressing this, we think that the authors should state this in the revised text.

Reviewer #3

(Remarks to the Author)

Reviewer #5

(Remarks to the Author)

The authors addressed my few reservations and the MS is overall improved. It can be published in its present form.

I only have one small comment concerning the efficacy of RNAi. The authors list a number of controls. While mCherry RNAi control for batch to batch effectiveness of RNAi treatment will tell you if RNAi plates and induction are OK, the HT115(DE3) bacteria can lose the genomic DE3 element (the source of the T7 polymerase). This can lead to a weak or absent RNAi phenotype. Chloroamphenicol is not systematically added in protocols and is not mentioned in M+M. Its absence could account for a lack of effect with *cec-4* and *lem-2*, for example, which don't have any obvious phenotype as read-out... This does not in anyway change the main conclusions of the papers.

Summary from the Authors

We thank the reviewers for their time and thoughtful comments. We have added significant new experiments to the manuscript as suggested by the reviewers, including more thorough evaluation of *eef-1A.1* and intronless reporter alleles. While these experiments did not change our conclusions, they add significant value to the manuscript and make the conclusions more complete. In regard to the overall style and our Discussion section, we have trimmed the manuscript down to be more concise, and we believe that the reviewers will notice that change. The Discussion in particular is now more focused and less speculative. During revision, a paper by Stewart et al. was published in Nature, and we have added it to the introduction, and changed our terminology from MAE to aRMAE to be consistent with it, and the most recent literature in humans.

Below are our responses to specific comments

Reviewer 1

“the antagonistic control of lifespan by the two histone methyltransferases. This part of the manuscript is a bit more difficult to understand and I would suggest the authors to better explain the relevance of these conclusions”

We revised the entire discussion section for clarity and removed redundant statements about the activity of the methyltransferases. We revised the model discussion and the

“I would suggest to split the conclusions in three parts: the mechanisms regulating MAE ; the epigenetic gambling and lifespan control ; the heritability.”

We took this into consideration and simply removed the heritability and epigenetic gambling from the discussion because they were distracting from the main points of the manuscript regarding the genes and domains controlling the silencing in the early embryo to regulate the monoallelic expression.

they reported a contradictory effect of *met-2* regarding what is known for transposons and heterochromatin but I would expect a development about this observation.

We removed this part of the discussion during revision because we agree that it was confusing to a reader. Within the main text and discussion, we reference Rechsteiner et. al., which showed that MET-2 and SET-25 regulate H3K9 methylation consistent with our model.

the same authors reported in 2021 (Nat. Com. 12(1), 6527) the importance of introns in controlling stochastic allele expression bias. I would expect a few words here related to this mechanism.

We thank the reviewer for the suggestion, and we have done these experiments. We tested the effects of *met-2* on the intronless versions of the *hsp-90* reporter and we have added that data to the manuscript in figure 2d, and in the revised text which now states, “To determine if either coding sequence or intron status was important for *met-2* to function as a negative regulator of aRMAE, we measured the full length *hsp-90T2A* alleles (different coding sequence than the promoter fusions) and *hsp-90* reporter alleles without introns in wild type and *met-2(null)* genetic backgrounds. In *met-2(null)* worms expressing the additional full length *hsp-90* coding sequence, intrinsic noise was significantly higher than in controls, shown in Figure 2c. Similarly, intronless *hsp-90* reporter alleles expressed in *met-2(null)* animals showed significantly higher intrinsic noise than controls, shown in Figure 2d. Taken together, we validated our initial screen result by determining that *met-2* negatively regulates aRMAE whether the reporters were inserted at different loci, had different coding sequences, or if the coding sequence contained introns or not.”

In addition to testing intronless alleles on *met-2* RNAi, we also tested the same animals on *set-25* RNAi, and found that SET-25 knockdown significantly changed the intronless phenotype to more BAE. This data is in Supplementary figure 3a. Moreover, we generated a *met-2* null allele in the intronless animals, and tested those animals on *set-25* RNAi. As with the intron-containing strain, the result was a BAE phenotype. This data is in Supplementary figure 3b. Overall, our data and conclusions are now in line with our 2021 paper and the new manuscript is much stronger as a whole.

they introduced some concepts or issues regarding penetrance, human diseases or longevity but the discussion is quite limited regarding these points. For example, the final conclusion refers to application in human health but it's not really supported by the manuscript. Consequently, the perspectives and applications of the results are not really clear and could have been better explained.

During the revision of this manuscript, Stewart et al reported in Nature monoallelic expression affecting human health. Additionally, we cite the work of our collaborator, Dr. Elizabeth Swisher, in identifying patients with endogenous silencing of *BRCA1* in tumors for ovarian cancer treatment. Our introduction and discussion sections have been narrowed, and both now connect the work here to the work in Stewart et al. We believe that the narrowed discussion section is more clearly written.

Reviewer 2 had no comments

Reviewer 3

1- The authors have shown the effect of MET-2 on MAE for a single-copy transgene driven by the promoter of *hsp-90*. It would be important to observe similar effects on the endogenous *hsp-90* gene. The authors have already generated such strain in Sands et al. 2021. Thus, they can study whether mutations in *met-2* and/or *set-25* affect the MAE of the endogenous *hsp-90* gene.

We agree with the reviewer that this would be the most useful additional data to support our conclusions. Unfortunately, we have found that homozygous edits of the *hsp-90* endogenous locus result in a developmental defect and arrest at L4 stage.

The strain referenced from our 2021 paper was a full length extra copy of *hsp-90* gene with a T2A and GFP/mCherry, not the native gene tagged with reporter alleles. We did add that T2A strain to the initial manuscript with the data for it in figure 2C.

Moreover, according to Sands et al. 2021, introns prevent MAE on the *hsp-90* gene and other transgenes. Thus, it would be interesting to test whether intron-less transgenes combined with *met-2* mutation show increased MAE.

We agree with the reviewer and have done these experiments, and the data are in the revised text, Figure 1d, Supplementary figure 3a,b. Please see response to reviewer 1 for complete revised text.

2- The authors showed that *eef-1A.1* transgenic reporter also shows some degree of MAE in the *met-2* mutant. However, they have not tested the effects of *lin-65* RNAi on MAE nor the impact of *set-25* and *lin-61* in suppressing MAE in *met-2* mutant. Testing these effects in *eef-1A.1* transgenic reporter would reinforce their conclusion on at least two transgenes (with different promoters).

We thank the reviewer for this suggestion as we have conducted these experiments and indeed the data make our conclusions far more robust. We have now tested the *eef-1A.1* reporter allele strain in both *met-2* and *set-25 RNAi* backgrounds. In addition, we made a *met-2 null* version of that strain, and tested it on *set-25 RNAi*. The data can be found in the main text Figure 3d, and in Supplementary Figure 3c,d.

3- In the introduction, the authors state that: “*met-2* mutant animals showed a pattern of monoallelic expression that was persistent throughout adulthood”. The authors have only measured MAE in one developmental time point (adulthood).

We thank the reviewer for noticing that we did not show the evidence for this. In this manuscript we are only measuring one time point, day 2 adult worms, which are in steady state gene expression if not perturbed. We have now added longitudinal experiments quantifying MAE in the exact same animals on different days of adulthood into Supplementary figure 10. In these studies, we measure MAE in the intestine of an anesthetized adult worm on an agar slide, then carefully put that worm back onto a culture plate before imaging again a couple days later. The procedure is delicate and time consuming and most worms do not orient their bodies on the slide identically at both time points, making it difficult to collect this data. Qualitatively, the cell-specific expression patterns and allele biases don't seem to change much in *most* animals intestine cells, even by days 8-12 of adulthood, in animals with strong reporter expression; rather the expression levels of many genes in worm intestine cells seem to

generally decline with age. In humans, the allele silencing also seems to last a lifetime, based on limited medical reports such as Okamoto et al.

Similarly, in the introduction, the authors state, "Reciprocal cross experiments showed that maternal MET-2 and SET-25 act in the E-cell of the 8-cell embryo to regulate MAE". This may be the case for some animals with fully monoallelic tissues. However, this is not always the case since there are animals with mosaic tissues, indicating that monoallelic expression was determined at later developmental stages. The proportion of these scenarios (mosaic vs monoallelic) should be quantified.

This is a very good point. It is true that we do not see complete aRMAE in every intestine tissue in every individual, indicating that aRMAE can be initiated early in the 8-cell embryo, but it can also be initiated later during development. For the manuscript revision, will we change the wording as the reviewer pointed out to reflect the fact that MAE is not always initiated in the 8-cell embryo. In addition, we added some qualitative assessments to give better context to the reader. Our new wording is as follows

"In *C. elegans*, the entire set of 20 intestine cells are descended from the single E-cell in the 8-cell embryo. Because we see persistent aRMAE throughout the entire intestine in some *met-2(null)* animals, it suggests that aRMAE can be initiated in the E-cell, propagated throughout mitotic divisions, and maintained into adulthood (e.g., Figs. 1a,d). Throughout this study, we observed that about 50% of *met-2(null)* animals had at least 50% or more of intestine cells in the same MAE state (~25% had almost or entirely monoallelic intestines), indicating that the silencing was most likely initiated in the E-cell of these animals. In the 8-cell embryo, there is very tight control of embryonic gene transcription⁶⁰. We hypothesized that because of the early initiation of aRMAE we (here), and others⁶¹ observe, and the rapid pace of *C. elegans* development, maternal or paternal proteins are responsible for initiation of aRMAE in the E-cell."

4- The authors infer the transcriptional regulation of MAE by MET-2 from the quantification of protein expression in the hsp-90 fluorescent reporter. However, no mRNAs or transcription ratio quantification of the reporter Cherry and GFP has been shown. Using smFISH will give the resolution to test the effect of *met-2* mutation in MAE at the transcriptional level.

We agree that we have not shown changes in transcript level. We do not plan to attempt to quantify smFISH at allele resolution because attempts at quantifying transcripts of individual alleles in animals show snapshots of bursting of individual alleles and not the average, both in our unpublished attempts and reported in *Drosophila* (Chen et al 2019), which we reference in our 2021 publication on introns and monoallelic expression. Thus, we have taken the approach of measuring the protein because, as reviewer 5 notes, and as in Chen et al 2019, transcriptional bursting of alleles uncouples the protein levels from the mRNA levels, and protein is the ultimate measure we are after for stable monoallelic expression at the protein

level. Transcripts of individual alleles have been studied in cell culture because cells in cell culture become permanently monoallelic. Organ-wide monoallelic expression can also be seen from bulk sequencing of biopsied human organs for the GTEx project, because bursting averages out from hundreds of cells to give an average bias per organ (reported in Kravitz et al 2023). Additionally, we have already completed analysis of human GTEx data and sequenced endometrial cancers, finding aRMAE to be prevalent in both healthy human tissues and in endometrial cancers (Manuscripts in preparation; Mariner et al. 2025; Jones et al., 2025).

5- The authors also aimed to test if MAE can be inherited. For this, they look at the allele-specific expression pattern in the progeny of met-2 mutant worms that showed mosaic transgene expression. They conclude that there is no inheritance of MAE because the progeny showed different allele-specific expression patterns than the parent, i.e., the parent had a mosaic intestine. In contrast, some progenies have a mostly monoallelic intestine (even though this was not quantified). The mosaic nature of monoallelic expression indicates that it is established randomly during development. Thus, it is unclear why the authors hypothesize that inheritance of MAE would imply inheritance of the pattern. To test if met-2 mutation could lead to epigenetic inheritance of monoallelic expression, the authors could cross out the met-2 mutation and assess if the resulting progeny still displays monoallelic expression.

We agree that the mix of monoallelic and mosaic expression indicates that it is established randomly during development and we are not trying to imply that there is any epigenetic inheritance – rather the opposite – in wild type or met-2 animals. We simply wanted direct evidence that the patterns are not inherited, which, though obvious to this reviewer, seemed important to show evidence for based on previous feedback. We actually did cross out the met-2 mutation in experiments where we show that met-2 acts in a maternal effect fashion in the early embryo, and those patterns are also not inherited in the F1, and the F2 progeny didn't seem very monoallelic under the stereoscope, indicating no epigenetic effect of general silencing – just the maternal effect – but we didn't quantify it because it wasn't a point of the manuscript. We hope our revised text indicates this. We did however quantify the number of animals that have an early silencing event, and thus, a nearly entirely monoallelic intestine, which we now include in the text when we discuss figure 4.

Minor comments:

1. The authors can shorten the introduction to improve the manuscript's readability.

We appreciate the suggestion and have done so.

2. The authors can combine some figure panels to facilitate the follow-up of the paper (and reduce the number of figures)

We have added some figure and text to the supplement.

3. The authors can perhaps substitute the name of the catalytic mutants with an acronym that facilitates the understanding of the catalytic mutation. For example, *met-2(wam406)* by *met-2cat*.

We have done this and indeed the manuscript is more readable with this change.

Reviewer 4 had no comments

Reviewer 5

1/ figure 4; rescue of *met-2* phenotype by *set-25* catalytic dead allele

I see no evidence of rescue at day 3, contrary to what is stated in the legend (panel c). If anything the double *set-25; met-2* is less fertile than the *met-2* single.

In panel (e) total fecundity in double mutants is less variable than in single *met-2*, but the median looks the same.

We thank the reviewer for their careful attention. We apologize for the distracting fecundity and lifespan section that took away from the focus on developmental genetic regulation of the monoallelic expression of the reporter alleles. The main point of that section was to show that the different alleles had different phenotypes for other traits besides aRMAE. We have moved that section to the supplementary information and corrected our erroneous text. We meant to state that the fertility of the *set-25(cat)* mutant is higher than wild type on day 4 – not that it rescues the fertility of the *met-2* mutant. We have corrected the supplementary text to state findings more clearly.

2/Given that the *met-2(wam007)* allele is more penetrant than the 406 allele, why did the authors not also test *set-25,met-2(wam007)* allele?

The *met-2(wam007)* allele was only more penetrant for fecundity and not significantly more penetrant for effects on MAE. In some cases we did test *set-25(RNAi)* on *met-2(wam007)* animals. See Fig 4 and 5 in main text and Supplementary figure 3. *set-25(RNAi)* animals showed similar MAE phenotypes to *set-25(cat)*, so we did not go ahead with making a traditional KO of *set-25*, as the point mutation was clearly a functional KO.

3/ could the authors show stats for significance on each panel rather than the legend ?

We will do this if the journal style allows it. We also have all detailed stats presented in the supplements.

4/ for the catalytically dead alleles, did the author perform WB analysis or IF to show expression of full-length protein and that catalytic activity is indeed knocked down? Although it is predicted that these point mutations only affect catalytic activity, this should be confirmed

We have sequenced the SET domain mutant animals and the genes are intact, so we believe there is no reasonable likelihood that expression has changed, given the number of generations since the genome edits, and the differences in lifespan phenotypes between the null and set alleles. We are trusting the extensive work of the Gasser lab on SET domain mutation validation here, which we have now referenced in the text. We have amended the text to more clearly state that the effects of our mutations on SET domain activity are reliant on the published works of others.

5/ Figure 4: efficacy of RNAi for genes that have no effect on MAE (cec-4, lem-2, lin-61, arle-1..) should be measured by RT-qPCR

We apologize for not clearly stating our controls, which we now do in the revised text. We did not use RT-PCR as a control for the RNAi screens. We did use other controls as follows. First, we sequenced all RNAi clones. Second, we utilized clones that had been reported by others to have known epigenetic phenotypes, for example lem-2 and cec-4, Towbin et al., 2012. Third, we used an mCherry RNAi to control for batch to batch effectiveness of RNAi treatment. Fourth, we observed phenotypes in important RNA clones. In addition, we validated our most important findings for met-2 and set-25 RNAi findings with CRISPR. Our methods section now reads,

“All RNAi clones were sequenced prior to seeding NGM plates. For RNAi experiments, we used only clones that had been reported to have effects (for example, cec-4 and lem-2 clones were used in Towbin et al., 2012). In addition, we used an mCherry expressing worm strain and anti-mCherry RNAi as an experimental control. For experiments, day 2 adult worms heterozygous for reporter alleles were placed on RNAi food or EV to lay eggs for two hours before the parents were removed. The process was repeated on day 4 for a total of two generations on RNAi food. On day eight from the start of the experiment, synchronous day 2 adults were harvested for microscopy. “

6) lifespan assays do not really contribute to an understanding of how met-2 and set-25 contribute to MAE

We agree that we did not explain that section of the manuscript well enough. These experiments were done just to show that the mutations had effects other than aRNAE, and to show that the domain and null mutants for met-2 had different phenotypes. We have since changed the format of the manuscript and placed this section in the supplements, and added a better explanation in the text. But for the reviewer, we should say that what we found important here was that total loss of met-2 has a different lifespan phenotype than met-2 with just a point mutation in its catalytic domain. This is similar to what was found by Susan Gasser with progeny. Our result just shows that the catalytically dead met-2 mutant lives longer than a met-2 null, distinguishing the two mutations by another phenotype in addition to the sequence validation shown in Supplemental Information.

In general, evidence linking germline immortality or longevity to somatic monoallelic expression is very weak. So although MET-2 and SET-25 are clearly involved in MAE, how MAE impacts animal physiology and the mechanism involved remains speculative.

We agree with the reviewer. We debated if we should even show any of the progeny or lifespan data because of the nebulous connection to MAE. We have decided to deemphasize it by putting the data in the Supplements and revising the text. We believe that the data has value for future experiments by us and others, but in the context of this manuscript, it does not play a strong role in our conclusions or working model.

Additional points

1/ In figure 3 it would help the reader to know the identity of the reporter alleles in each of the panels a-d. This is only explicitly stated for panel D in the legend (mTagBFP2 or mNeptune expression). The color of each fluorochrome should also be given: eg scarlet/violet

We corrected this.

2/ The authors suggest that increased interindividual variation in progeny production of met-2 mutants (Figure 6f) may reflect monoallelic expression as a form of « epigenetic gambling » with some individuals having abundant progeny and others very few. While this may well be an explanation, inactivation of chromatin factors in *C. elegans* often results in reduced brood sizes, with decreases that can vary greatly between individuals

We agree with the reviewer that random chromatin factor inactivation can reduce brood sizes variably due to the variation in the silencing of the chromatin. However, these results have moved to the Supplement as they were a bit of a distraction from the main point of the paper, the developmental genetic regulation of aRMAE in a model metazoan. We really want to keep the focus on aRMAE regulation because it seems to be prime time to study this in both models and humans with the new tools we have. While this paper was under revision, a new paper from Stewart et al came out in *Nature*, showing how MAE causes differences in immune cell functions in humans. Since this manuscript was submitted, we have been engaged with the human GTEx data, and sequencing of endometrial cancers, validating what we have found in *C. elegans*, in terms of seeing monoallelic expression in whole tissues and tumors.

Reviewer #1 (Remarks to the Author):

This revised manuscript presents now a compelling study of autosomal random monoallelic expression (aRMAE) in *C. elegans*, focusing on the antagonistic roles of the H3K9 histone methyltransferases MET-2 and SET-25. The authors have convincingly demonstrated that maternal MET-2 activity limits SET-25-driven silencing in early embryogenesis, thereby establishing persistent monoallelic expression patterns in the intestine. Compared to the initial version, the authors have responded accurately to the reviewers' comments. Moreover, they have added new data, with inclusion of intronless hsp-90 reporters and eef-1A.1 reporter. The overall conclusions are now more solid. The text has been substantially improved, mainly throughout a more focused and tightened discussion. Speculative sections on lifespan, immortality and epigenetic gambling have been deleted or diminished, and the text now centers on the mechanistic antagonism between MET-2 and SET-25 and its developmental timing. The authors have also changed some wording and concepts to refer to very recent literature. They are now using the term aRMAE instead of the more general MAE, which soundly refers to Stewart et al., 2025. The new figure 7 graphically summarizes their model of monoallelic regulation and is good visual synthesis. In conclusion, the revised version represents a significant improvement over the original. It delivers a clear mechanistic insight into how opposing histone methyltransferases establish and maintain aRMAE in *C. elegans*. The authors' rebuttal and the additional data address the reviewers' concerns and significantly increase the robustness of the conclusions. The manuscript, in its current form, is suitable for publication.

We thank the reviewer for their comments and the resulting improvements to the manuscript.

Reviewer #2 (Remarks to the Author):

We thank the authors for adequately addressing most of our concerns. We still believe that one limitation of the study is that it does not directly assess transcription as a read-out of monoallelic stochastic expression. Even though we acknowledge that methodological limitations prevent the authors from addressing this, we think that the authors should state this in the revised text.

We thank the reviewer for helping us frame the technical limitations of our study. We now more clearly state that we do not assess transcription as a readout of monoallelic expression in the manuscript. In the revised introduction, text now reads: "We note that, unlike probe or sequencing-based approaches, our approach with fluorescent reporter alleles does not quantify the transcripts of each allele of a gene, but

instead quantifies the resulting protein products of each allele.”

Reviewer #3 (Remarks to the Author):

Reviewer #5 (Remarks to the Author):

The authors addressed my few reservations and the MS is overall improved. It can be published in its present form.

I only have one small comment concerning the efficacy of RNAi. The authors list a number of controls. While mCherry RNAi control for batch to batch effectiveness of RNAi treatment will tell you if RNAi plates and induction are OK, the HT115(DE3) bacteria can lose the genomic DE3 element (the source of the T7 polymerase). This can lead to a weak or absent RNAi phenotype. Chloramphenicol is not systematically added in protocols and its not mentioned in M+M. Its absence could account for a lack of effect with *cec-4* and *lem-2*, for example, which don't have any obvious phenotype as read-out...

This does not in anyway change the main conclusions of the papers.

We thank the reviewer for this technically sharp observation and now note in the methods that our frozen bacterial RNAi stocks were pulled from media containing chloramphenicol, but not subsequently cultured in its presence to ensure other scientists are aware that we did not continue with chloramphenicol selection after taking the initial stock from its selection media. The methods state that we use carbenicillin, tetracycline and IPTG in the media we use to grow our plasmid-sequence confirmed RNAi clones.

The revised text in the methods now reads:

“RNAi stocks were pulled from stock media containing ampicillin, tetracycline and chloramphenicol.”